# A transcriptome atlas of leg muscles from healthy human volunteers reveals molecular and cellular signatures associated with muscle location

**Tooba Abbassi-Daloii[1]†, Salma el Abdellaoui[1], Lenard M Voortman[2], Thom TJ Veeger[3], Davy Cats[4], Hailiang Mei[4], Duncan E Meuffels[5], Ewoud van Arkel[6], Peter AC 't Hoen[1,7]\*, Hermien E Kan[3,8]\*, Vered Raz[1]\***

[1]Department of Human Genetics, Leiden University Medical Center, Leiden, Netherlands; [2]Division of Cell and Chemical Biology, Leiden University Medical Center, Leiden, Netherlands; [3]C.J. Gorter MRI Center, Department of Radiology, Leiden University Medical Center, Leiden, Netherlands; [4]Sequencing Analysis Support Core, Leiden University Medical Center, Leiden, Netherlands; [5]Orthopedic and Sport Medicine Department, Erasmus MC, University Medical Center Rotterdam, Rotterdam, Netherlands; [6]Orthopedics, Medisch Centrum Haaglanden, Den Haag, Netherlands; [7]Centre for Molecular and Biomolecular Informatics, Radboud Institute for Molecular Life Sciences, Radboud University Medical Center, Radboud, Netherlands; [8]Duchenne Center Netherlands, Leiden, Netherlands

**\*For correspondence:**
peter-bram.thoen@radboudumc.nl (PAC'tH);
H.E.Kan@lumc.nl (HEK);
v.raz@lumc.nl (VR)

**Present address:** †Department of Bioinformatics, Maastricht University, Maastricht, Netherlands

**Competing interest:** The authors declare that no competing interests exist.

**Abstract** Skeletal muscles support the stability and mobility of the skeleton but differ in biomechanical properties and physiological functions. The intrinsic factors that regulate muscle-specific characteristics are poorly understood. To study these, we constructed a large atlas of RNA-seq profiles from six leg muscles and two locations from one muscle, using biopsies from 20 healthy young males. We identified differential expression patterns and cellular composition across the seven tissues using three bioinformatics approaches confirmed by large-scale newly developed quantitative immune-histology procedures. With all three procedures, the muscle samples clustered into three groups congruent with their anatomical location. Concomitant with genes marking oxidative metabolism, genes marking fast- or slow-twitch myofibers differed between the three groups. The groups of muscles with higher expression of slow-twitch genes were enriched in endothelial cells and showed higher capillary content. In addition, expression profiles of Homeobox (*HOX*) transcription factors differed between the three groups and were confirmed by spatial RNA hybridization. We created an open-source graphical interface to explore and visualize the leg muscle atlas (https://tabbassidaloii.shinyapps.io/muscleAtlasShinyApp/). Our study reveals the molecular specialization of human leg muscles, and provides a novel resource to study muscle-specific molecular features, which could be linked with (patho)physiological processes.

## Editor's evaluation

Skeletal muscle groups varied in biomechanical and can show a diffential involvement pattren in muscular disorders. Molecular characteristics of leg muscles from healthy young human males is presented in this study. It provides the first comparitive of gene signatures that will be valuable to understand muscle involvement in normal and abnormal physiological and pathological conditions.

## Introduction

Skeletal muscles have *grosso modo* similar functions, generate the force for mobility and skeleton support, and maintain the body homeostasis. However, skeletal muscles differ in biomechanical and physiological features. These features include the size and contractile properties of the motor units and myofibers, differences in shortening velocity, resistance to fatigue, and differences in innervation and perfusion (*Valentine, 2017*). Yet, the molecular and cellular differences that contribute to this muscle specialization are not fully understood. A molecular atlas for different skeletal muscles could assist in deciphering the molecular basis of muscle-specific physiological features. Such an atlas may also be used to study differential muscle involvement in various conditions, such as muscular dystrophies, myopathies, differences in regenerative potential, physiological compensation in sports and sarcopenia.

During aging and in muscle diseases, some muscles are affected earlier than others. This is often referred as muscle involvement pattern, which can be characteristic of a given muscular pathology (*Carlier et al., 2011*; *Raz et al., 2015*; *Albayda et al., 2018*; *Brogna et al., 2018*; *Diaz-Manera et al., 2018*; *Servián-Morilla et al., 2020*). Several studies suggested that muscle-specific intrinsic molecular factors may explain this muscle involvement pattern (*Kang et al., 2005*; *Rahimov et al., 2012*; *Huovinen et al., 2015*; *Raz et al., 2016*; *Terry et al., 2018*; *Hettige et al., 2020*; *Xi et al., 2020*). For example, differences in the cellular pathways and myofiber type (slow- and fast-twitch myofibers) composition between muscles could play a role (*De Micheli et al., 2020*; *Rubenstein et al., 2020*; *Xi et al., 2020*), but may not fully explain the muscle involvement patterns.

Most of the studies characterizing the molecular variation between muscles were performed in mice (*Campbell et al., 2001*; *Porter et al., 2001*; *Haslett et al., 2005*; *von der Hagen et al., 2005*; *Raz et al., 2018*; *Terry et al., 2018*; *Hettige et al., 2020*), where muscle-specific mRNA profiles were linked to distinct myofiber type composition (*Campbell et al., 2001*; *Raz et al., 2018*; *Hettige et al., 2020*).

Since human muscle-related pathologies are not always recapitulated in mouse models (*van Putten et al., 2020*), understanding molecular variations between skeletal muscles should be performed in human samples. Only a few studies compared mRNA profiles between muscles from healthy human adults, and these studies face several limitations. Skeletal muscles are highly affected by age (*McCormick and Vasilaki, 2018*; *Aversa et al., 2019*), yet, the age range in previous studies was broad (*Kang et al., 2005*; *Huovinen et al., 2015*). Moreover, the number of sampled muscles and subjects were limited (*Abbassi-Daloii et al., 2020*). A study using postmortem material (*Kang et al., 2005*) only partly reflects molecular composition in living muscles due to storage in cooling conditions. Understanding muscle involvement in different pathologies can benefit from a molecular atlas of human muscles.

We generated a transcriptome atlas from six leg muscles and two locations from one muscle to explore molecular variations within and between muscles. Paired samples were obtained from 20 healthy male subjects of 25±3.6 years old. We show that the seven muscle tissues clustered into three groups, distinguished by cell type composition and mRNA expression profiles. We confirmed the transcriptome analyses with large-scale quantitative immunohistochemistry and RNA in situ hybridization procedures. We discuss the value of this skeletal muscle atlas resource to understand human health and pathologies affecting skeletal muscle tissues.

## Results

### Transcriptome atlas of adult human skeletal muscles

To determine molecular signatures marking leg muscles, we generated a transcriptome atlas of human skeletal muscles by sequencing biopsies from five upper leg muscles, gracilis (GR), semitendinosus (ST), rectus femoris (RF), vastus lateralis (VL), and vastus medialis (VM) muscles, and one lower leg muscle, gastrocnemius lateralis (GL; *Figure 1A*). We also investigated molecular differences within one muscle by including biopsies from the middle and distal end of the semitendinosus muscle (STM and STD, respectively). These two biopsies were treated as independent muscle samples in subsequent analyses (*Figure 1B and C*). In total, 128 samples from 20 individuals (aged 25±3.6 yr) were analyzed (*Figure 1—figure supplement 1*), making this currently the largest freely available human muscle atlas. *Supplementary file 1* shows the sample characteristics.

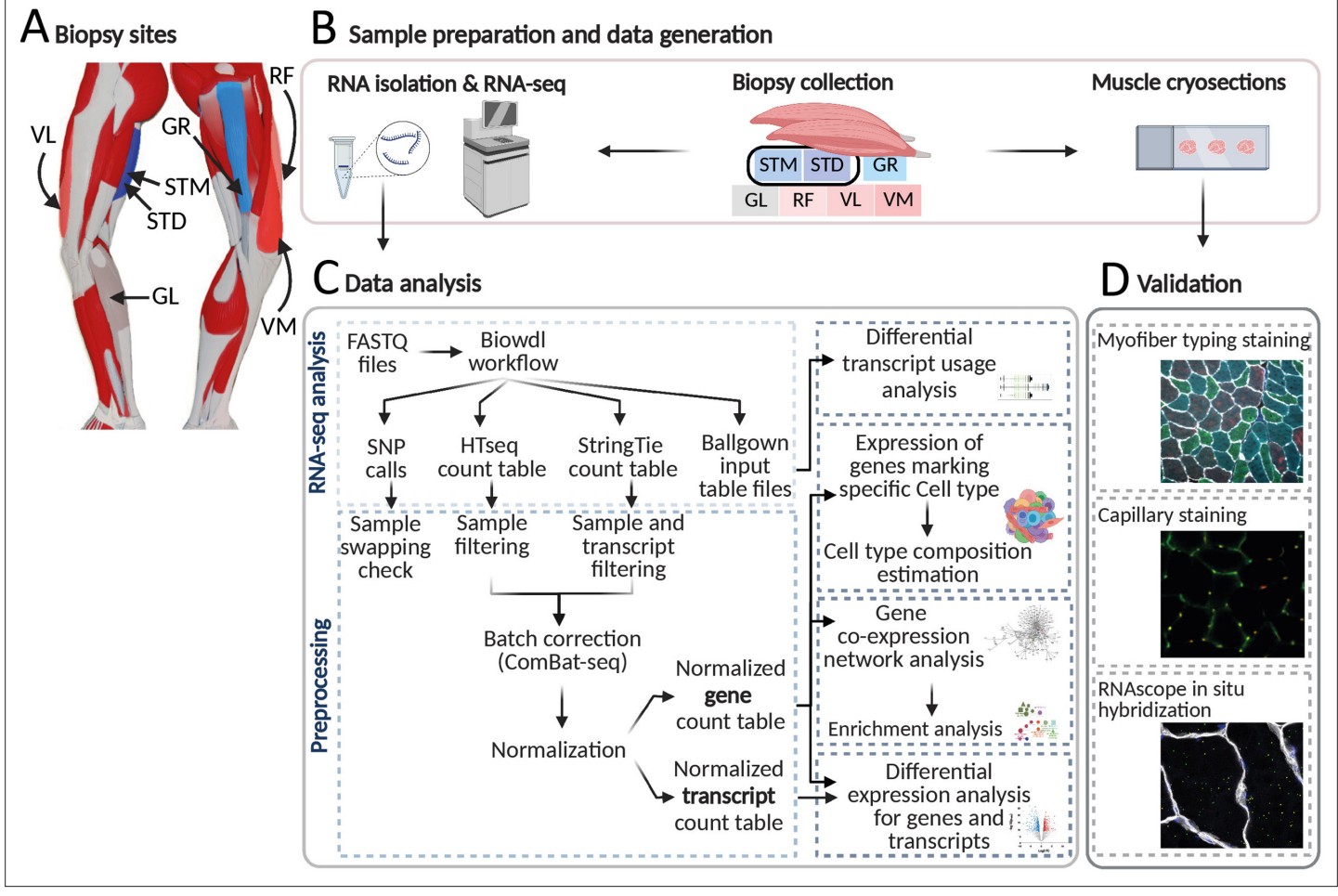

**Figure 1.** An overview of biopsies' location and the study workflow. (**A**) A schematic overview of the leg muscles. Arrows point to the muscles that were included in this study. The biopsies, with exception of STM (semitendinosus-middle), were taken from the distal area. (**B–D**) The study overview includes cryosectioning, RNA-isolation and sequencing (**B**) data analysis (**C**) and validations (**D**). Created with BioRender.

The online version of this article includes the following figure supplement(s) for figure 1:

**Figure supplement 1.** Analysis framework.

**Figure supplement 2.** RNA isolation protocols.

**Figure supplement 3.** The overview of RNA-seq samples.

**Figure supplement 4.** The quality control and batch correction of RNA-seq data.

## Variation in cell type composition between different muscles

Skeletal muscle is a heterogeneous tissue containing multiple cell types. The differences in the abundance of these cell types can be reflected in bulk RNA-seq profiles. Therefore, we used RNA-seq data to first explore possible cell type heterogeneity between leg muscles. We summarized the expression level of genes marking each cell type present in human skeletal muscles by calculating their first principal components (eigenvectors; *Supplementary file 2*). We used the eigenvalues of the eigenvectors representing the different cell types to cluster the muscles (*Figure 2A*) and to identify cell types with significant differences in relative abundance between muscles (*Figure 2—figure supplement 1*). The muscle tissues clustered into three groups, Group 1 (G1): GR, STM, and STD; Group 2 (G2): RF, VL, and VM; GL was the only muscle in Group 3 (G3) (*Figure 2A*). The relative abundance of endothelial cells was statistically the most different between muscles, with higher abundance in G2 and G3 than in G1 (*Figure 2A–B*, *Figure 2—figure supplement 1*). Other cell types marking blood vessels, namely pericytes, post-capillary venule (PCV) endothelial cells, natural killer (NK) cells, T and B cells, and myeloid cells, clustered together with the endothelial cells and all showed a higher abundance in G2 and G3

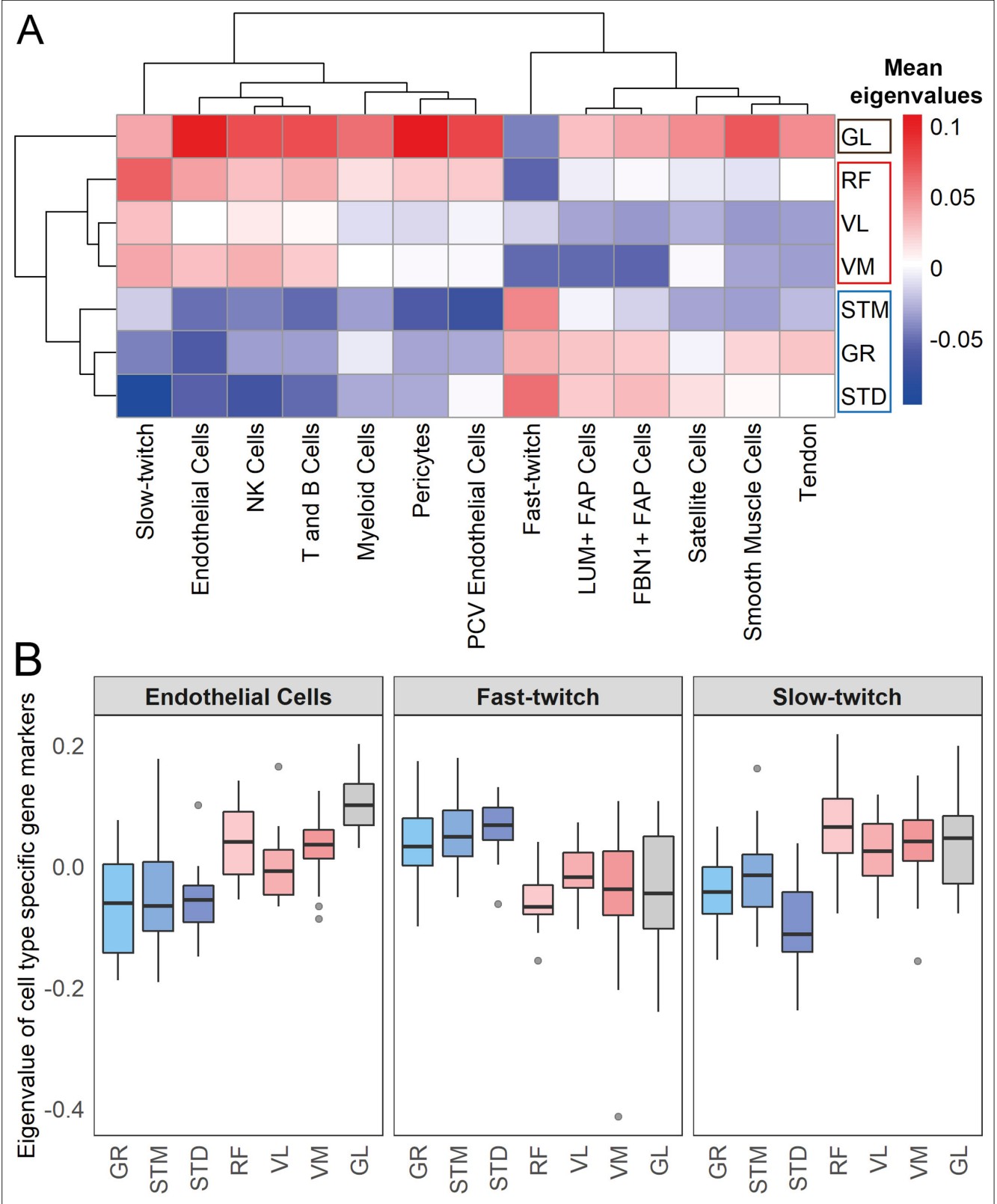

**Figure 2.** Muscles cluster into three main groups based on cell type composition. (**A**) The heatmap shows the mean eigenvalues of genes marking each cell type across all the individuals. Each row shows a muscle, and each column shows a cell type. FAP stands for fibro-adipogenic progenitors. (**B**) The boxplot shows the eigenvalues for the endothelial cells, fast-twitch, and slow-twitch myofibers per muscle. The boxes reflect the median and interquartile range.

*Figure 2 continued on next page*

*Figure 2 continued*

The online version of this article includes the following figure supplement(s) for figure 2:

**Figure supplement 1.** Cell type composition pairwise comparison between muscles.

**Figure supplement 2.** Cell types' eigenvalues.

compared with G1 (*Figure 2A–B*). These results could suggest a higher capillary density and blood perfusion in/of the muscles in G2 and G3.

Genes marking fast-twitch myofibers showed overall higher expression levels in G1, while slow-twitch genes were higher in G2 and G3 (*Figure 2A–B*). Differences in the relative abundance of non-muscle cell types, pericytes, immune cells, and endothelial cells, distinguished G3 from the G1 and G2 muscles (*Figure 2—figure supplement 2*, *Figure 2A–B*).

While STM and STD showed significant differences in the relative abundance of genes marking endothelial cells (higher expression in STD) and slow-twitch myofibers (higher expression in STM), there were no significant differences between ST and GR, and within the G2 muscles. This suggests that differences between regions of the same muscle may be larger than differences between distinct muscles (*Figure 2—figure supplement 1*).

## Further study of differences in myofiber type composition between groups of muscles

To confirm the differences in myofiber types between muscles, we performed immunofluorescence staining for all muscles with a mixture of antibodies to three MyHC isoforms and anti-laminin antibody (*Figure 3A*). We developed a semi-automated image processing workflow to segment the myofibers using laminin staining and to quantify the fluorescence intensity of each MyHC isoform per myofiber. We next identified myofiber types by clustering all the myofibers using the MFI values of the three MyHC isoforms. The vast majority of the myofibers (94%) were assigned to three major clusters (*Figure 3B*). Each myofiber cluster had a major MyHC isoform (*Figure 3C*). Consistent with our study in human *vastus lateralis* muscle (*Raz et al., 2020*), the results here suggest that the myofibers are generally not purely type -I, -IIA, or -IIX but contain a mix of myosin heavy chain isoforms. We observed relatively high correlations from 0.55 to 0.62 between normalized gene expression of the dominant MyHC in each cluster and the proportion of myofibers assigned to the corresponding cluster (*Figure 3D–F*, *Figure 3—figure supplement 1A*). This correlation demonstrates the reliability of our RNA-seq-based assessment of MyHC expression.

In agreement with the results of the RNA-seq cell type composition analysis (*Figure 2A–B*), the quantitative histology analysis demonstrated a higher proportion of slow-twitch (oxidative) myofibers and a lower proportion of MyHC2X-dominated myofibers in G2 and GL (G3) muscles than in G1 muscles (*Figure 3G*, *Figure 3—figure supplement 1B*). The quantitative histology analysis further showed that G2 muscles had a higher proportion of MyHC2A-dominated myofibers than the G3 muscle (*Figure 3G*, *Figure 3—figure supplement 1B*), highlighting a distinct myofiber type composition of the GL muscle.

The myofiber composition results further showed a higher proportion of MyHC2X-dominated myofibers in STD than in STM, whereas STM had a higher proportion of MyHC1 and MyHC2A-dominated myofibers (*Figure 3—figure supplement 1B*). This confirms the existence of regional differences within a muscle (*Bindellini et al., 2021*).

## Higher capillary density in GL

The RNA-seq cell type composition analysis suggested a higher proportion of endothelial and other cell types marking blood vessels in the GL muscle than in other muscles. To confirm this observation, we immunostained the endothelial cells using antibodies against Endoglin (ENG) and CD31 proteins (*Tey et al., 2019*). We included cryosections of GL and STM with the largest differences in the expression of genes marking endothelial cells (*Figure 4A*). We observed a higher proportion of CD31-positive areas in GL (*Figure 4B*), which was consistent with higher *CD31* RNA expression levels in this muscle (*Figure 4C*).

Next, we determined the muscles' capillary density by counting small circular objects stained positive for both CD31 and ENG (*Wehrhan et al., 2011*). We observed a higher capillary density in GL

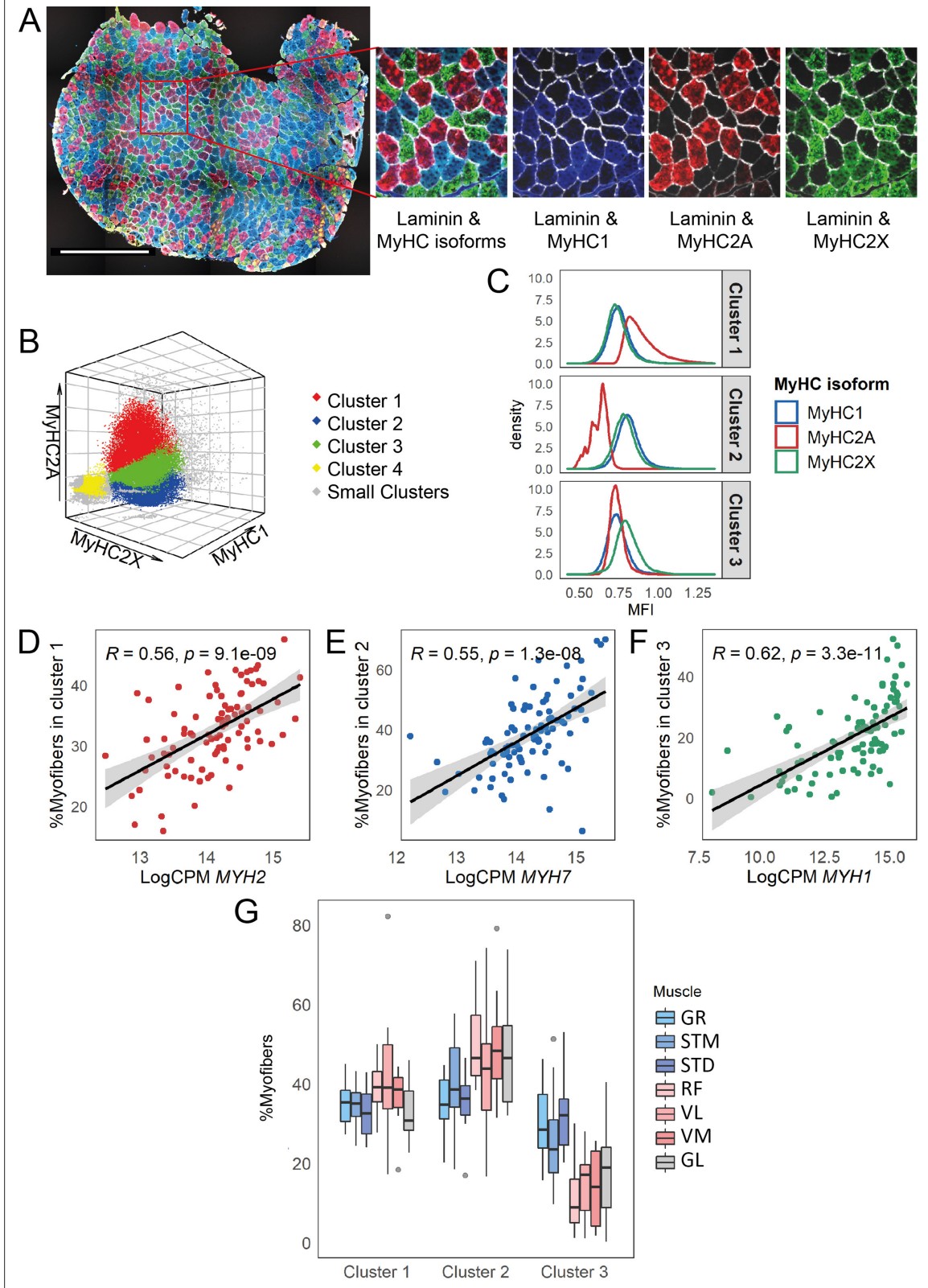

**Figure 3.** Myofiber type composition is consistent with the expression level of genes marking fast and slow-twitch myofibers. (**A**) A representative immunostaining image. The overlay of each myosin heavy chain isoform and laminin are shown separately. Scale bar is 1000 μm. (**B**) A 3-D scatterplot of the MyHC isoforms MFI. Each dot represents a myofiber. Myofibers in the three largest clusters are denoted with red (Cluster 1), blue (Cluster 2), and green (Cluster 3). The objects with low MFI values for all the isoforms are denoted in yellow (Cluster 4, ~2% of all the dots). In gray are ~4% of

*Figure 3 continued on next page*

*Figure 3 continued*

myofibers assigned to many small clusters. (**C**) Density plots show MFI distribution for each MyHC isoform in the three largest clusters. MFI values are scaled (without centering) and transformed. (**D–F**) Scatterplots show the proportion of the assigned myofibers to each of the largest clusters and the normalized expression of the gene coding the isoform with a relatively higher expression in that specific myofiber cluster. (**G**) The boxplot shows the proportion of myofibers in the three largest clusters per muscle. Each muscle is depicted with a different color, with G1 muscles in blue, G2 muscles in red and the G3 muscle in gray. The boxes reflect the median and interquartile range.

The online version of this article includes the following figure supplement(s) for figure 3:

**Figure supplement 1.** The association of myofiber clusters with MyHC expression, and myofiber composition differences between muscles.

compared with STM (*Figure 4D*). This observation is consistent with a higher proportion of endothelial cell types in GL compared with muscles in G1 or G2.

## Gene and transcript expression profiles and molecular pathways distinguishing muscle clusters

We next investigated whether muscle-specific gene and transcript expression profiles, not explained by cell type composition, could also be found in our dataset. To this end, we determined the differentially expressed genes (DEGs; *Figure 5—figure supplement 1A*, *Supplementary file 3*) and the differentially expressed transcripts (DETs; *Figure 5—figure supplement 1B*, *Supplementary file 4*) for every pairwise comparison. DEGs that were driven by differences in cell type composition were excluded (Pearson's $R > 0.5$ between gene expression levels and the eigenvector of any cell type). Similarly, the transcripts originating from genes related to differences in cell type composition were excluded from the list of DETs. The proportion of DEGs and DETs that were not driven by cell type composition but discriminated each pair of muscles are shown in *Figure 5A* and *Figure 5B*, respectively. The muscles clustered in a similar way as was observed in the cell type composition analysis: GR, STM, and STD (G1), RF, VL, and VM (G2), and GL (G3) (*Figure 5A–B*).

We then investigated the transcript usage differences in each pair of muscles using the IsoformSwitchAnalyzeR algorithm (*Vitting-Seerup and Sandelin, 2019*). We found a total of 79 genes with at least one transcript usage difference in any pairwise comparison of muscles (*Figure 5—figure supplement 1C*, *Supplementary file 5*). Out of 79 significant genes, 14 were excluded as they were related to the differences in cell type composition (*Figure 5—figure supplement 2*). *Figure 5C* shows the number of genes with significant transcript usage differences in each pair of muscles. The limited number of genes with significant transcript usage differences compare to DEGs and DETs analyses could be, because we could not account for the individual effect in this analysis in the same as for the DEG and DET analyses.

To further study muscle-specific expression profiles, we applied weighted gene co-expression network analysis (WGCNA). We identified 34 modules of co-expressed genes (*Supplementary file 6*). For each module, we calculated the module eigengene (ME) that represents gene expression levels of the genes in the module. We then implemented a pairwise comparison to find modules showing significant differences in every pairwise comparison (*Figure 5—figure supplement 1D*, *Figure 5—figure supplement 3A*). Out of the 34 modules, 27 showed a difference between at least two muscles (module size range: 38–1459; containing 10,695 genes in total). Nine out of the 27 muscle-related modules had at least five genes marking a specific cell type and were therefore defined as modules driven by differences in cell type composition and were not considered for further analysis (*Figure 5—figure supplement 3B*). *Figure 5D* shows the remaining modules that were not driven by cell type composition, nevertheless distinguished pairs of muscles. We then plotted the mean eigenvalues of muscle-related modules in a heatmap (*Figure 5E*) to determine the clustering of muscles based on the expression patterns of genes in the modules. The WGCNA-based clustering was consistent with the cell type composition and differential expression groups (*Figure 5E*). In total, 7 out of the 18 muscle-related modules demonstrated higher expression levels in G1, four modules had higher expression in G2 and G3, and three modules demonstrated higher expression levels in G3 only (*Figure 5E*). In addition, although none of the modules showed distinct expression patterns between muscles in G2 and between ST and GR, M.21 module showed higher expression levels in STM than STD (*Figure 5—figure supplement 3A*).

To explore the molecular and cellular pathways in the three groups, functional enrichment analysis was performed in the muscle-related modules. The most significantly enriched biological processes

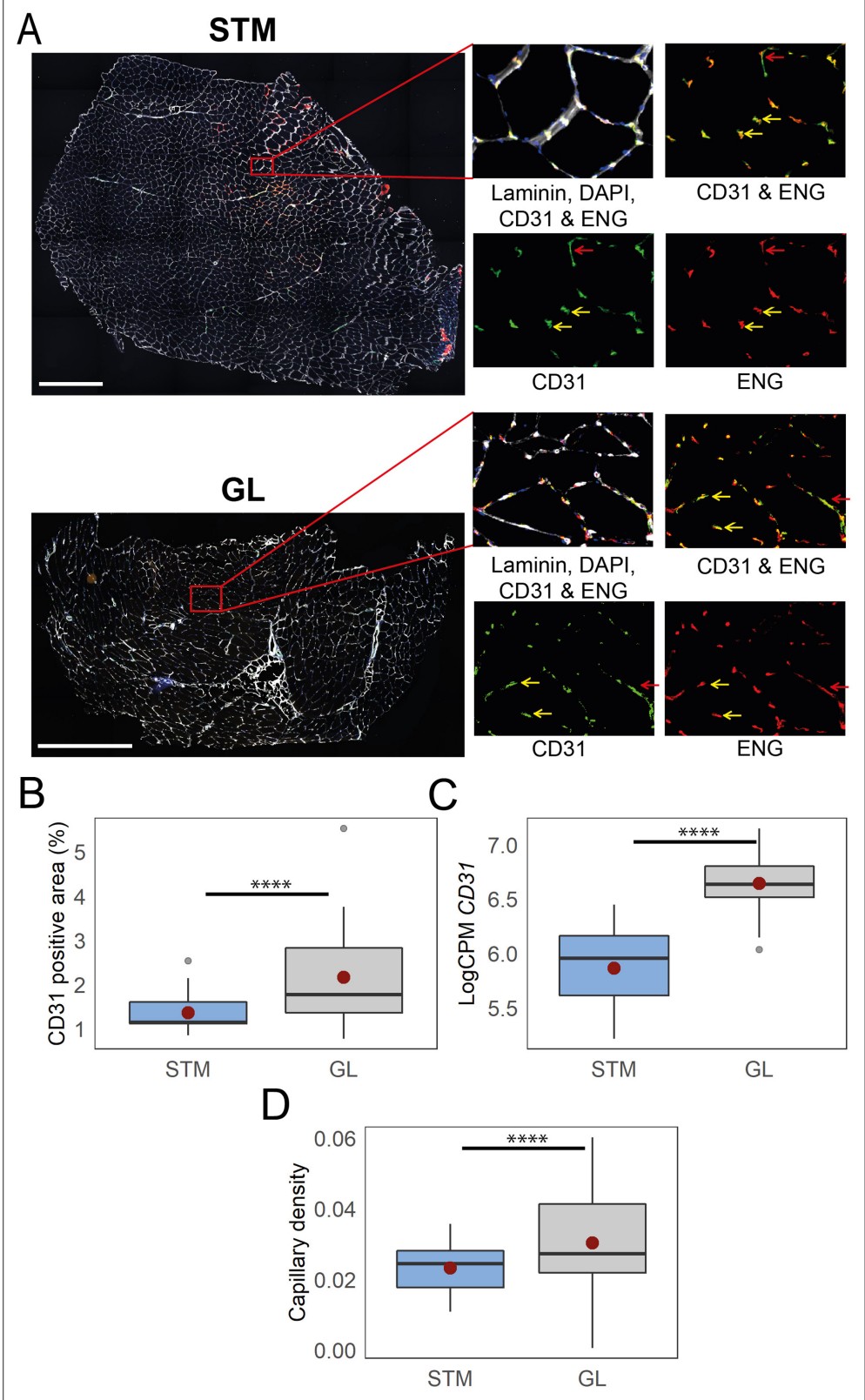

**Figure 4.** Immunostaining confirms higher capillary density in GL compared with STM muscles. **A)** Representative images of STM and GL cross-sections immunostained with CD31 (green), ENG (red), and laminin (white). An enlargement of the boxed region is shown on the right: merged and separate channels (CD31 and ENG). Examples of objects recognized as capillaries are depicted by yellow arrows. Examples of objects that were not

*Figure 4 continued*

considered as capillaries due to circularity values are shown by red arrows. Scale bar 1000 μm. (**B**) The box plot shows the percentage of CD31 positive area in the two muscles. (**C**) The box plot shows the normalized expression of *CD31* gene in the two muscles. (**D**) The boxplot shows the estimated capillary density in the two muscles. The capillary density was defined as the number of objects (3–51 μm$^2$) with an overlap between CD31 and ENG per unit cross-sectional area of the muscle. The boxes reflect the median and interquartile range (N=19 per muscle). The red dots on the boxes show the mean. **** p-value $<1 \times 10^{-6}$ (linear mixed-model).

and molecular functions within these modules are listed in *Table 1* (a complete list is in *Supplementary file 6*).

## Higher expression of mitochondrial genes in G2 and G3 muscles consistent with higher proportion of slow myofibers

In the M.13 module, with higher expression in G2 (VL, VM, and RF) and G3 (GL), the mitochondrial-related genes were enriched (*Table 1*). Eighteen out of 122 genes enriched for mitochondria in this module were hub genes, highly interconnected genes in the module (*Table 1*). To assess a potential impact on mitochondrial metabolic processes, we mapped the 122 genes to mitochondrial pathways (*Figure 6*). The most enriched processes were the respiratory electron transport chain in oxidative phosphorylation, the tricarboxylic acid (TCA) cycle, and beta-oxidation. This observation suggests a higher oxidative metabolism in G2 and G3, which is consistent with a higher proportion of slow myofibers.

## Homeobox transcription factors contribute to the mRNA diversity between the three groups of muscles

An enrichment for 'anterior/posterior pattern specification' in M.14 was observed, with higher expression in G2 and G3 (*Table 1*). This module included *HOX* hub genes (*Figure 7A*). To assess whether the diversity between the groups of muscles was associated with the pattern of *HOX* gene expression, we plotted the normalized expression of all expressed *HOX* genes across all samples (*Figure 7B*). Remarkably, clustering based on *HOX* gene expression clearly separated the G1 from the G2 and G3 muscles (*Figure 7B*). Additionally, we observed that VL muscle sub-clustered apart from the two other muscles in G2 (*Figure 7B*). Moreover, 11 out of 36 *HOX* genes were assigned to three of the muscle-related modules (M.14, M.30, and M.32), which showed the largest differences between the three groups of muscles (*Figure 7B*). From each of these three muscle-related modules, one *HOX* gene was selected (*HOXA11*, *HOXA10* and *HOXC10*) to further confirm the differences in expression between muscles using the in situ hybridization (ISH) procedure (*Figure 8A*). We included samples from GL and STM showing the largest difference in *HOX* gene expressions. *HOXA11* was excluded from quantification analysis as it showed a low signal-to-noise ratio, in line with a lower expression level of RNA-seq-based assessment of *HOXA11* compare to *HOXA10* and *HOXC10*. The *HOX* signal was mainly localized in myofibers (*Figure 8A*). Per sample, the average number of foci per myofiber was calculated revealing a higher number of *HOXA10* and *HOXC10* single molecule RNAs in STM compared with GL (*Figure 8B*). The ISH results were consistent with the RNA-seq data (*Figure 8B–C*), further demonstrating the robustness of our RNA-seq data.

## Web application for exploring transcriptome atlas of human skeletal muscles

To facilitate data reuse and exploration of the human skeletal muscle atlas, we developed a web application (https://tabbassidaloii.shinyapps.io/muscleAtlasShinyApp), enabling users to look up the sample information and the expression of any gene of interest. In addition, users can explore the list of genes used for the cell type composition analysis and their expression levels across all the samples. Furthermore, users can list and visualize the differentially expressed genes and the modules and their hub genes.

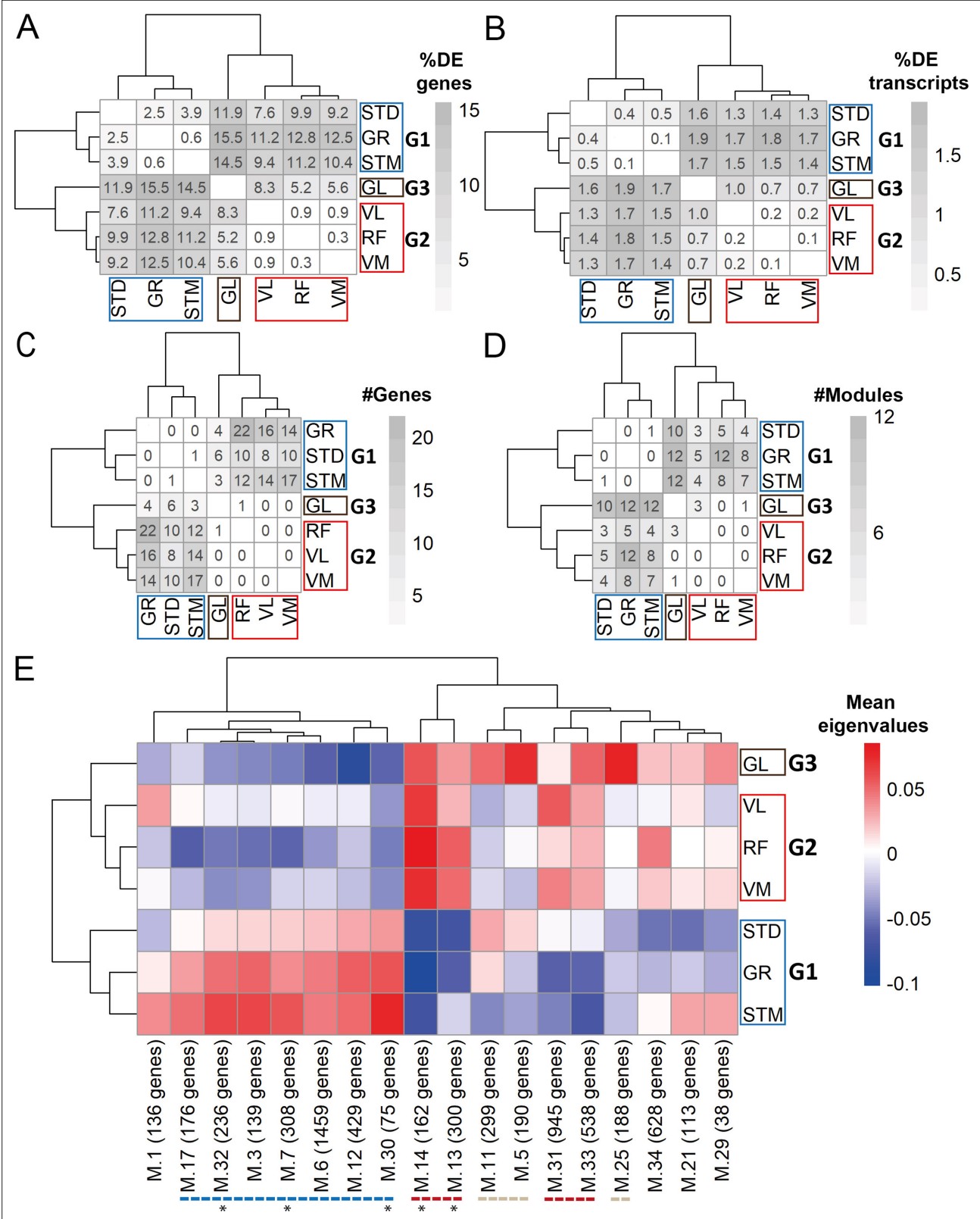

**Figure 5.** Gene expression differences between three groups of muscles not driven by cell type composition. **A)** Symmetric heatmap plot shows the percentage of DEGs in different pairwise comparisons. Genes with a high Pearson correlation (*R*>0.5) with the eigenvector of any cell type (cell type related genes) are excluded. (**B**) Symmetric heatmap plots show the percentage of DETs in pairwise comparisons. Transcripts originating from cell type related genes are excluded. (**C**) as in (**B**), but for the number of genes having at least a significant transcripts usage difference. (**D**) Symmetric heatmap

*Figure 5 continued on next page*

*Figure 5 continued*

plot shows the number of modules that were not driven by cell type composition and were significantly different in each pairwise comparison. Each row or column in **A–D**) represents a muscle. (**E**) The heatmap shows the modules that reflect the intrinsic differences between groups of muscles. Each row represents a muscle, and each column shows a muscle-related module that was not driven by cell type composition. Color-coded cells show the corresponding average of eigenvalues across all individuals (N=20). Modules with an overall higher expression in G1 or G3 are underlined by a blue or gray dashed line, respectively. The red dashed line underlines the modules with an overall higher expression in both G2 and G3. The black asterisks show modules with the largest differences between the three groups of muscles.

The online version of this article includes the following figure supplement(s) for figure 5:

**Figure supplement 1.** DEA and WGCNA also clustered muscles in three groups.

**Figure supplement 2.** Overview of genes with significant transcript usage difference.

**Figure supplement 3.** WGCNA modules.

## Discussion

We generated a large skeletal muscle transcriptome atlas from 20 young healthy males. We included six leg muscles and two locations within one muscle. The atlas presented in this study is unique in terms of the number of muscles, the individuals included, and the age range of the participants. We confirmed the RNA-seq analysis using large-scale quantitative immunohistochemistry and mRNA in situ hybridization. Based on cell type composition, differential gene and transcript expression analyses, differential transcript usage analysis and WGCNA, the seven leg muscle tissues consistently clustered into three groups: (G1) GR, STM, and STD; (G2) VL, VM, and RF; (G3) GL. The muscles in G2 and G3 (VL, VM, RF and GL) showed higher proportions of slow myofiber types and higher capillary densities. GL, the only lower leg muscle, was distinct from VL, VM, and RF in its lower proportion of type 2 A myofibers and a higher proportion of non-muscle cells. *HOXA10* and *HOXC10* expressions were lower in VL, VM, RF and GL than in GR, STM, and STD muscles.

### Molecular diversity between muscles in different anatomical locations

The muscles included in this study mobilize and stabilize the knee joint. Muscles of the hamstrings (ST and GR) clustered together (G1), and muscles of the quadriceps (RF, VL and VM) clustered together (G2). The Hamstrings and quadriceps alternate in contraction and relaxation to flex, extend and stabilize the knee and aid in moving the thigh. GL, the only lower leg muscle in our set, is in the posterior side of the leg, allowing flexion of the knee and plantar flexion of the ankle. Our study suggests that there is little molecular diversity between muscles of the same group, as compared to muscles in different groups of muscles.

We observed a higher proportion of fast-twitch myofibers in G1 compared with G2 and G3. This could be due to the role of the hamstrings in activities that require a large power output since fast-twitch myofibers are used more in these activities than slow-twitch myofibers (*Bottinelli et al., 1999*; *Willigenburg et al., 2014*, *Camic et al., 2015*). Slow-twitch myofibers have a higher mitochondrial content compared with fast-twitch myofibers (*Berchtold et al., 2000*; *Gouspillou et al., 2014*). Consistently, G2 and G3 muscles showed higher expression of genes encoding for mitochondrial proteins and a higher ratio of slow-twitch myofibers compared with G1. Slow-twitch muscles are also supplied by a denser capillary network (*Nishiyama, 1965*; *Murakami et al., 2010*; *Korthuis, 2011*). Indeed, we observed a higher capillary density and higher endothelial cells in G3. The three groups of muscles also differed by the expression of *HOX* genes, specifically, *HOXA* and *HOXC* family members. *Hox* genes establish the anterior/posterior patterning during vertebrate embryonic limb development (*Zakany and Duboule, 2007*). Interestingly, the development of these groups of leg muscles differs in developmental time (*Diogo et al., 2019*), consistent with the expression of *Hox* genes (*Zakany and Duboule, 2007*). *Hox* genes expression is not limited to embryonic development, but was found also in adult mouse muscles (*Houghton and Rosenthal, 1999*; *Yoshioka et al., 2021*), and *Hoxa10* gene was differentially expressed across adult limb mouse muscles (*Yoshioka et al., 2021*). Moreover, *Yoshioka et al., 2021* demonstrated that *Hoxa10* expression in adult satellite cells affects muscle regeneration in mice. Here, we show that both *HOXA* and *HOXC* gene families are expressed in myofibers, and their expression levels differ between leg muscles. *Yoshioka et al., 2021* also showed the expression of *HOX* genes in adult human muscle tissues. Yet, *Terry et al., 2018* concluded that the expression pattern of *Hox* genes in adult muscles is insufficient to explain the mRNA expression diversity in adult

**Table 1.** Top enrichment results of muscle-related modules not driven by cell type composition.

| Module | Term | FDR | #Enriched genes | Hub genes |
|---|---|---|---|---|
| **Higher expression in G1 (GR, STM, and STD)** | | | | |
| M.30 (75 genes) | IRE1-mediated unfolded protein response | $4\times10^{-4}$ | 4 | |
| M.32 (236 genes) | Hormone-mediated signaling pathway | $9\times10^{-3}$ | 11 | CARM1, WBP2, ZBTB7A |
| M.7 (308 genes) | Negative regulation of nucleobase-containing compound metabolic process | $6\times10^{-4}$ | 51 | CREBBP, DAB2IP, FOXK2, LARP1, RXRA, THRA |
| | Chromatin organization | $1\times10^{-2}$ | 27 | ARID1B, CREBBP, HUWE1 |
| M.17 (176 genes) | Chromatin modifying enzymes | $4\times10^{-3}$ | 11 | HCFC1, SETD1A |
| **Higher expression in G2 (RF, VL, and VM) and G3 (GL)** | | | | |
| M.31 (945 genes) | RNA splicing | $6\times10^{-5}$ | 54 | SNRNP70 |
| | Histone modification | $3\times10^{-3}$ | 47 | KAT2A |
| M.33 (538 genes) | Apical junction complex | $2.4\times10^{-2}$ | 11 | MICALL2 |
| M.13 (300 genes) | Mitochondrion | $4\times10^{-45}$ | 122 | AIFM1, ATP5F1A, ATP5F1B, CKMT2, COQ9, DLD, DLST, FH, GHITM, HADHA, HADHB, IMMT, MFN2, NDUFS2, PDHA1, PDHB, TRAP1, UQCRC2 |
| M.14 (162 genes) | Anterior/posterior pattern specification | $4\times10^{-3}$ | 7 | HOXA11 |
| **Higher expression in G3 (GL)** | | | | |
| M.5 (190 genes) | Regulation of lipid metabolic process | $8\times10^{-4}$ | 15 | ADIPOQ, ADRA2A, CIDEA, LEP, LGALS12, PDE3B, SCD |
| M.25 (188 genes) | Ameboidal-type cell migration | $5\times10^{-3}$ | 15 | CFL1, PML, TGFB1 |
| | Positive regulation of muscle cell differentiation | $9\times10^{-3}$ | 6 | EHD2, ENG, NIBAN2, TGFB1 |
| M.11 (299 genes) | Golgi membrane | $2\times10^{-3}$ | 30 | ASAP2, MAN1A1 |
| | Regulation of nervous system development | $8\times10^{-3}$ | 30 | IQGAP1 |

mouse skeletal muscles. Whether *HOX* genes are transcriptionally active in adult myofibers is a subject for future studies.

## Potential relevance to muscle disease and aging

In several muscle-related diseases like muscular dystrophies (MDs), muscle weakness and pathological features like replacement of muscle tissue with fat starts in specific muscles and spreads to others as the disease progresses (*Emery, 2002*). This pattern differs between diseases, and the reason for the disease-specific involvement pattern is unknown. Exploring the molecular signatures that contribute to the differences between muscles may elucidate the pathophysiology of these diseases. For example, in Duchenne muscular dystrophy (DMD), which is caused by mutations in the *DMD* gene, the quadriceps is involved earlier, whereas the hamstring muscles are less involved, and the GR is spared (*Wokke et al., 2014*; *Hooijmans et al., 2017*). The observed higher expression level of the *DMD* gene in ST and GR may be related to the late involvement in DMD patients during disease progression.

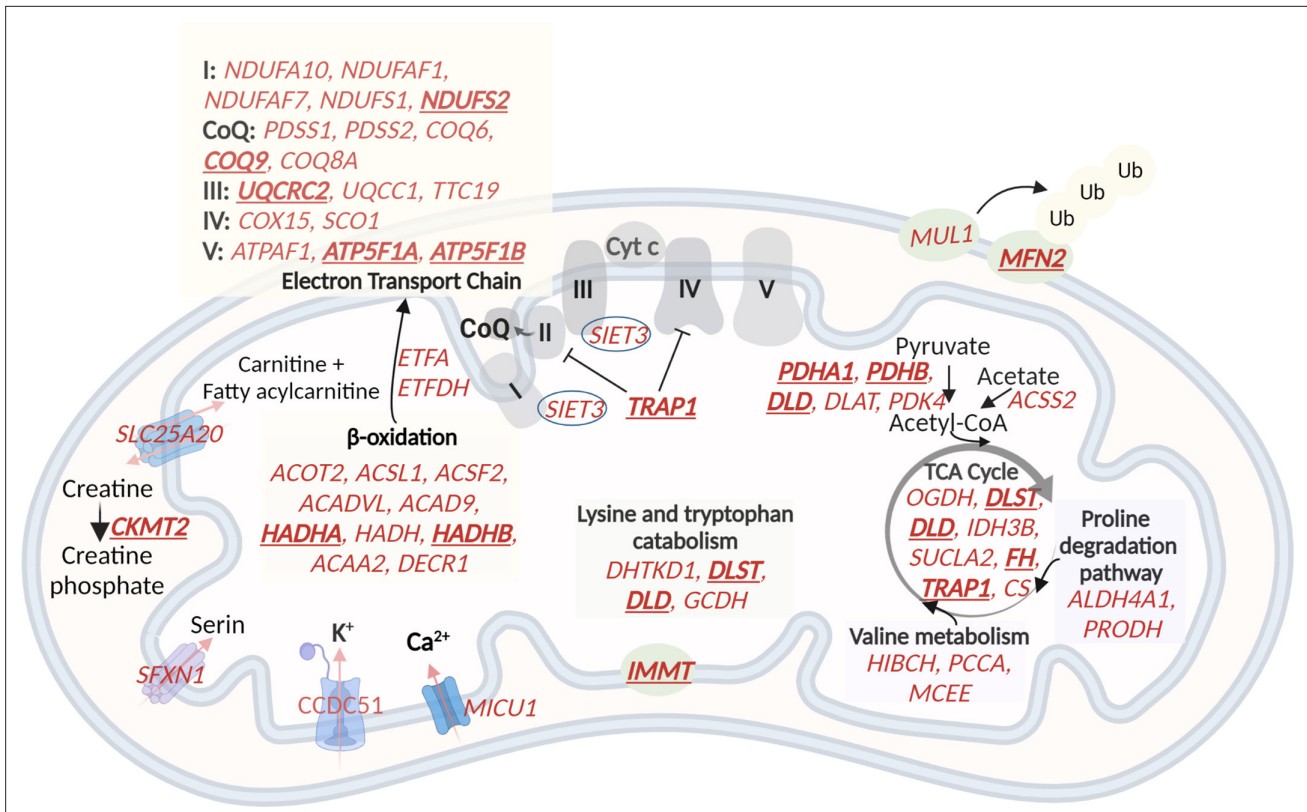

**Figure 6.** A schematic representation of genes with higher expression in G2 and G3, related to oxidative phosphorylation and metabolic pathways in the mitochondria. 60 (out of the 122) mitochondrial genes with higher expression in G2 and G3 are shown in red. The electron transport chain, lysin and tryptophan catabolism, TCA cycle, and beta-oxidation are shown. The hub genes are underlined and in bold. Created with BioRender.

Accessing the expression level of genes and implementing a quantitative approach (*Veeger et al., 2021*) to evaluate the association between leg muscle architectural characteristics and gene expression levels could be performed for other muscle diseases.

## Regional differences within muscles

The molecular and cellular differences between the samples from distal and middle locations of ST were larger than the differences between ST and GR (*Figure 2—figure supplement 1*, *Figure 2—figure supplement 2*). One module of co-expressed genes, M.21, showed a different expression pattern between STM and STD. This module was enriched for the cellular amino acid catabolic process and monocarboxylic acid catabolic process (*Supplementary file 7*). While the distal side of the ST muscle has a rounded tendon, the differences between STM and STD cannot be explained by contamination of tendon tissue or closer proximity to the tendon, because we did not find a difference in the estimated tenocyte proportions between biopsies collected from the distal and middle parts of the muscle (*Figure 2—figure supplement 1*, *Figure 2—figure supplement 2*). The myofiber composition was different between the distal and medial parts of the ST muscle (*Figure 2—figure supplement 1*, *Figure 3—figure supplement 1B*). A divergent myofiber type composition of biopsies from superficial and deep areas of the same human muscle was reported by *Johnson et al., 1973* for the GL, RF, VL, VM, *adductor magnus*, *soleus*, and *tibialis anterior* muscles in the leg and thigh. *Bindellini et al., 2021* also reported different proportions of MyHC2A myofibers in distal and middle parts of *tibialis anterior* in mice.

## Interindividual differences were larger than differences between muscles

Despite the narrow age range and an inclusion of only one gender in our study, we observed that the percentage of variance explained by the individual surpassed the variance explained by the muscles

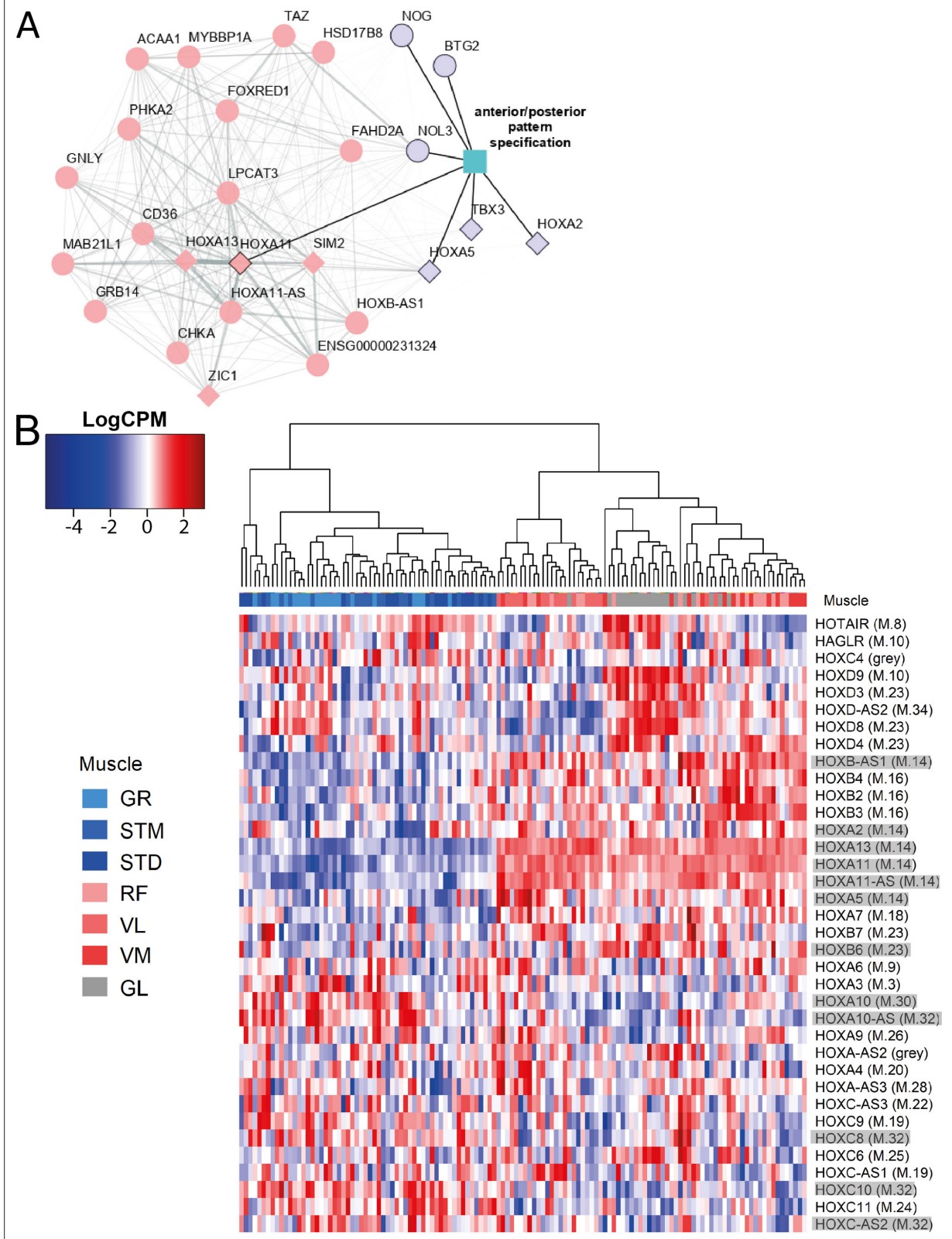

**Figure 7.** The expression patterns of *HOX* genes cluster muscles in the same groups. **A**) The graph shows the co-expression subnetwork of *HOX* genes and genes related to anterior/posterior pattern specification assigned to the M.14 module. Diamonds indicate transcription factors while other genes are indicated by circles. Pink and purple nodes represent the hub genes and non-hub genes, respectively. The genes related to anterior/posterior pattern specification have a black border. The edge thickness reflects the degree of topological overlap. Topological overlap is defined as

*Figure 7 continued on next page*

*Figure 7 continued*

a similarity measure between each pair of genes in relation to all other genes in the network. High topological overlaps indicate that genes share the same neighbors in the co-expression network. (**B**) Normalized expression of all HOX genes (scaled by row) represented as a heatmap. The hierarchical clustering was generated using the normalized expression values. Each row represents a gene and each column represents a sample. The side color of columns indicates different muscles. The module in which the gene assigned is given between parentheses. Eleven highlighted HOX genes are assigned into muscle-related modules which showed the largest differences between the groups of muscles (M.14, M.30, and M.32).

(*Figure 1—figure supplement 4D*). This is in agreement with findings from *Kang et al., 2005*. The inter-individual variations are possibly resulting from genetic and environmental (activity, exercise, diet, etc.) factors. To account for inter-individual variation, we included the individual as a random effect in the statistical models in cell type and differential expression analyses. In the WGCNA, we constructed a consensus gene co-expression network by merging the co-expression networks separately constructed per individual. In the downstream steps of the WGCNA workflow, the same as in cell type analysis and differential expression analyses, individuals were included as a random effect in the models. Only after properly accounting for interindividual differences, we could identify the intrinsic differences between leg muscles.

## Study limitations

Differences in cell type composition between muscles are best captured using single-cell sequencing. Previous single-cell (*De Micheli et al., 2020*; *Rubenstein et al., 2020*; *Xi et al., 2020*) and single nucleus (*Orchard et al., 2021*; *Perez et al., 2021*) studies reported the cellular composition of adult human muscles, where single nucleus profiling is preferred because myofibers cannot be dispersed into single cell suspensions. The high costs associated with single-cell technologies are currently prohibitive for performing large scale analyses of >100 samples such as performed in our study. Here, we evaluated differences in cellular composition by deconvolution of bulk RNA-seq based on marker genes reported in single-cell studies. This approach appeared to be suitable for analyzing differences in cellular composition between large sets of samples, as we observed good consistency with immunohistochemistry-based analyses of myofiber type and endothelial cell composition. A limitation of the deconvolution approach is, however, that this only captures cell types for which discriminative marker genes are available.

We further acknowledge that RNA expression levels do not necessarily match protein abundance in muscles and do not reflect post-translational modifications (*Greenbaum et al., 2003*; *Liu et al., 2016*). Although a protein atlas could relate to muscle cell function better than RNA expression profiles, generating a genome-wide proteome in skeletal muscles is challenging, as muscle proteomes are dominated by the high abundance of high-molecular-weight sarcomeric proteins, and capturing the low abundance proteins is challenging. Despite this limitation, we showed consistency between results obtained by mRNA expression profiling and immunohistochemical staining of the proteins that were in focus in our study.

In summary, we demonstrated divergent molecular and cellular compositions between skeletal muscles in different anatomically adjacent locations. Overall, the consistency of the gene expression patterns, and the results obtained from the immunohistochemistry and RNA in situ hybridization experiments indicate the high accuracy and reliability of the transcriptome atlas generated in this study. Therefore, this atlas provides a resource for exploring the molecular characteristics of muscles and studying the association between molecular signatures, muscle (patho)physiology and biomechanics.

## Materials and methods
### Subject characteristics and biopsy collection

Healthy male subjects (aged 18–32) undergoing surgery of the knee for anterior cruciate ligament (ACL) reconstruction using hamstring autografts were recruited from outpatient clinics of two hospitals: Erasmus Medical Center and Medisch Centrum Haaglanden. Inclusion criteria included age, sex, and the amount of routine exercise. Subjects eligible for reconstructive ACL surgery were mobile, had full range of knee motion, minimal to no knee swelling and had physiotherapy until the surgery.

A total of seven biopsies were taken from six different leg muscles (*Figure 1A*). To study molecular differences within the muscle, two biopsies from the middle and distal sides of the semitendinosus

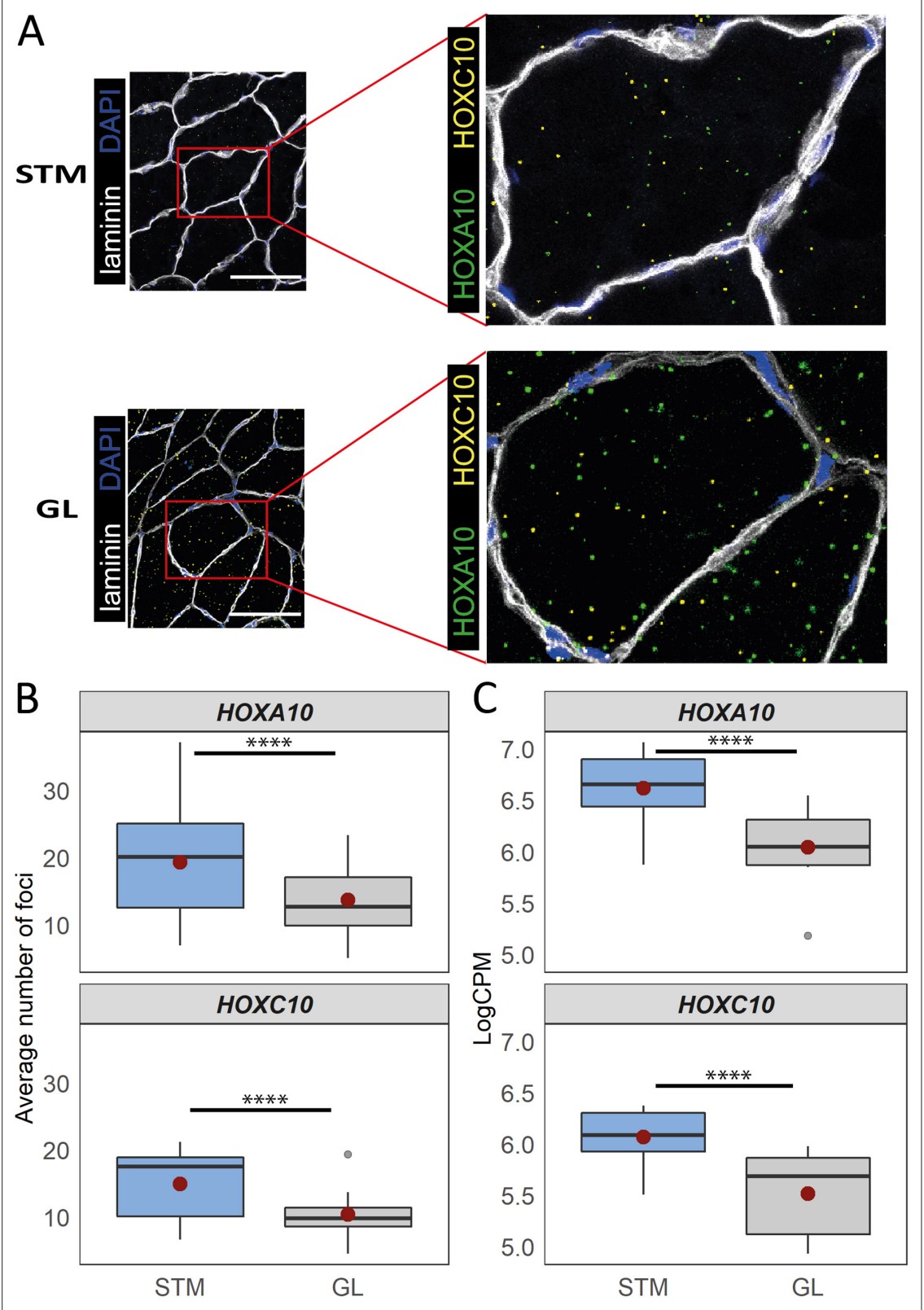

**Figure 8.** Distinct expression of *HOX* genes confirmed by RNAscope. (**A**) Representative in situ hybridization images of *HOXC10* and *HOXA10* in cryosections of STM and GL. The image is a merge image of the channels used for laminin, DAPI, *HOXC10* and *HOXA10* staining. Scale bar is 100 µm. (**B**) The boxplots show the average number of foci per myofiber (y-axis) in STM and GL muscles (x-axis). (**C**) The boxplots show the normalized

*Figure 8 continued on next page*

*Figure 8 continued*

expression of *HOXA10* (top) and *HOXC10* (bottom) in STM and GL muscles. The boxes reflect the median and interquartile range (N = 12 per muscle). The red dots on the boxes show the mean. **** p-*value* < 1 × 10⁻⁶ (linear mixed-model).

The online version of this article includes the following figure supplement(s) for figure 8:

**Figure supplement 1.** A representative image of negative control probes in the RNAscope experiment in STM muscle cryosections.

muscle (STM and STD, respectively) were collected. During the surgery, the tendons of the gracilis (GR) and semitendinosus muscles were used to reconstruct the ACL, and biopsies from these muscles were taken directly from the graft after harvesting the autografts at the beginning of the operation. After the ACL construction, biopsies from gastrocnemius lateralis (GL) rectus femoris (RF), vastus lateralis (VL), and vastus medialis (VM) muscles were taken by percutaneous biopsy (modified *Bergstrom, 1975*) using a minimally invasive biopsy needle. All biopsies were immediately frozen in liquid nitrogen and were kept at –80 °C.

The study was approved by the local Medical Ethical Review Board of The Hague Zuid-West and the Erasmus Medical Centre and conducted in accordance with the ethical standards stated in the 1964 Declaration of Helsinki and its later amendments (ABR number: NL54081.098.16). All subjects provided written informed consent prior to participation.

## Sample processing, RNA isolation, and cDNA library preparation

Biopsies were cryosectioned for RNA isolation, immunofluorescence staining, and in situ hybridization. For each sample, three cryosections of 16 μm thick were collected onto SuperFrost slides (Thermo Fisher Scientific, 12372098) and stored at −20 °C prior to staining. For in situ hybridization, the cryosections were mounted on SuperFrost Plus Adhesion slides (Thermo Fisher Scientific, 12625336) and stored at −80 °C. For the RNA isolation, cryosections were transferred into MagNA lyser green beads tubes (Roche, 3358941001). Then, they were homogenized in QIAzol lysis reagent (Qiagen, 79306) using the MagNA Lyser. Subsequently, total RNA was purified with chloroform. For samples from a subset of individuals, RNA was precipitated with isopropyl alcohol (*Supplementary file 1*). For the other samples total RNA was mixed with an equal volume of 70% ethanol and further purified with miRNeasy Mini Kit (217004, Qiagen) using the manufacturer's protocol (*Supplementary file 1*). To evaluate the effect of two different RNA isolation protocols, RNA from five GR samples were isolated with both protocols (*Figure 1—figure supplement 2*). For both protocols, DNA was removed using RNAse-free DNAse set (Qiagen, 79254) using the manufacturer's protocol. RNA integrity was assessed with the Agilent 2100 Bioanalyzer using Eukaryote Total RNA Nano chips according to the manufacturer's protocol (Agilent BioAnalyzer, 824.070.709) (*Supplementary file 1*).

Poly(A) library preparation was performed in four batches each with 39 samples at Leiden Genome Technology Center (LGTC, the Netherlands). Information on the RNA isolation protocol and library preparation batches used for each sample can be found in *Supplementary file 1*. Samples from different muscles and individuals were equally distributed in each library batch to minimize a batch effect bias. Approximately 200 ng of total RNA was used as starting material. mRNA was enriched using oligo dT beads (polyA +bead-based enrichment), fragmented, and converted to cDNA using random hexamers and SuperScript III (Invitrogen). End-repair, A-tailing, and adapter ligation were performed using NEBNext chemistry (New England Biolabs) and xGen dual index UMI adapters (Integrated DNA Technologies) according to the manufacturer's protocol. Finally, USER digest (New England Biolabs) and 15 cycles of library amplification were performed. Libraries were purified with XP beads and analyzed for size and purity on a Bioanalyzer DNA HS chip (Agilent BioAnalyzer, 5067–1504).

## Bulk RNA-sequencing and analysis

Illumina sequencing was performed by GenomeScan BV (Leiden, the Netherlands) on a Novaseq-6000 producing paired-end 2×150 bp reads. Fastq files were processed using the BioWDL pipeline for processing RNA-seq data (v3.0.0, https://zenodo.org/record/3713261#.X4GpD2MzYck) developed by the sequencing analysis support core (SASC) team at LUMC. The BioWDL pipeline performs FASTQ pre-processing, RNA-seq alignment, deduplication using unique molecular identifiers (UMIs), variant calling, and read quantification. FastQC (v0.11.7; ) (https://www.bioinformatics.babraham.ac.uk/projects/fastqc/) was used for checking raw read QC. Adapter clipping was performed using Cutadapt

(v2.4; *Martin, 2011*) with default settings, followed by checking the QC using FastQC. RNA-Seq reads' alignment was performed using STAR (v2.7.3a; *Dobin et al., 2013*) against the GRCh38 reference genome. PCR duplications were removed based on UMIs using UMI-tools (v0.5.5; *Smith et al., 2017*). Gene read quantification was performed using HTSeq-count (v0.11.2; *Anders et al., 2015*). The expression of known transcripts were quantified using StringTie (3.7.3)(*Pertea et al., 2015*). Ensembl version 98 (http://sep2019.archive.ensembl.org/) was used for genes and transcripts annotation. Samples with less than 5 M reads assigned to annotated exons were re-sequenced or excluded from all downstream analyses. A SNP calling was performed using GATK4 (v4.1.0.0; *McKenna et al., 2010*). Possible sample swapping was checked using an SNP panel with 50 SNPs (*Yousefi et al., 2018*). The similarity for calls of these SNPs showed that two samples in the same RNA isolation batch were swapped. We revised the labels of these two samples in our dataset for downstream analyses.

We performed all the analyses in RStudio Software (v1.3.959; *RStudio-Team, 2020*) using R Statistical Software (v4.0.2) (*R Development Core Team, 2020*). Samples with more than 5 M reads assigned to annotated exons were included in all downstream analyses (*Figure 1—figure supplement 3*). The HTSeq count table was used to create a DGEList object using the edgeR Bioconductor package (v3.30.3) (*Robinson et al., 2010*). The filterByExpr function from the edgeR Bioconductor package was used to keep genes with 10 or more reads in at least 16 samples (the number of samples in the smallest muscle group). The dataset was normalized using the calcNormFactors function (considering trimmed mean of M-values [TMM] method) from the edgeR Bioconductor package.

## Quality control and batch effect correction

We performed principal component analysis (PCA) to evaluate the main difference between samples in an unsupervised manner. Log-transformed expression values after normalization by counts per million (CPM) were used to calculate principal components using the base function prcomp with the center and scale argument set to TRUE.

We then performed the analysis of variance to determine the factors driving gene expression variations. We estimated the contribution of known biological (muscle tissues and Individuals) and technical (RNA isolation protocol, RIN score, initial RNA concentration, library preparation batch, sequencing lane, and library size) factors on variation of gene expression. Data transformed by the *voom* function from the limma Bioconductor package (v3.44.3; *Law et al., 2014*; *Ritchie et al., 2015*) was used to fit a linear model for each gene. We included all biological and technical factors as fixed effects and fitted the following linear model:

$$voom-transformed\,expression_{gene}\,muscle+individual+RNA\,isolation\,protocol+RIN\,score+concentration+$$

$$library\,preparation\,batch+sequencing\,lane+library$$

We used ANOVA from the car R package (v3.0–10; *Fox and Weisberg, 2019*) to estimate the relative contribution of each of these factors in the total variation of gene expression. Outcomes from both PCA and ANOVA revealed a strong library preparation batch effect (*Figure 1—figure supplement 4A* and C), while the effect of other technical factors (RNA isolation protocol, initial RNA concentration, RIN score, and library size) was minimal (*Figure 1—figure supplement 4C*). Accordingly, the HTSeq count table was corrected for the batch effect by the ComBat-seq Bioconductor package (*Zhang et al., 2020*). The muscle was included in the ComBat-seq model to preserve possible molecular differences between muscles. The ComBat-Seq count table was used to create a DGEList object, followed by removing the low expressed genes and additional normalization using filterByExpr and calcNormFactors functions, respectively.

Similarly, the batch effect was corrected for the StringTie transcript count table using the same ComBat-seq's model. Next, a DGEList object was generated and the low expressed transcripts were filtered considering the same threshold used for gene dataset. The transcripts were further filtered to only include the transcripts of genes that were present in the filtered gene dataset. Finally, the transcript dataset was normalized with TMM method.

Outcomes of PCA and ANOVA confirmed the proper removal of the batch effect (*Figure 1—figure supplement 4B* and D). In addition, the percentage of variance explained by the individual was found to be bigger than the variance explained by the muscle (*Figure 1—figure supplement 4D*). We, therefore, included the individual as a random effect in all different analyses. Moreover, the RIN score

was not considered as an exclusion criterion as it did not contribute to gene expression variation (*Figure 1—figure supplement 4D*).

## Cell type composition estimation

We collected lists of genes marking different cell types that are present in human skeletal muscles from different studies (*Smith et al., 2013*; *Kendal et al., 2019*; *Perucca Orfei et al., 2019*; *Rubenstein et al., 2020*; *Supplementary file 2*). The expression of genes marking each cell type was summarized by their eigenvector (first principal component). We subsequently fitted a linear-mixed model to the eigenvector of each cell type using the lmer function from the lmerTest R package (3.1–3; *Kuznetsova et al., 2017*). These models included muscle as a fixed effect and individual as a random effect shown in the formula below:

$$eigenvector_{celltype}muscle + (1 \vee individual) + error$$

We tested the significance of fixed effects with the ANOVA from the car R package. The Benjamini-Hochberg false-discovery rate (FDR) was applied to adjust for multiple testing. We conducted post-hoc pairwise comparisons using the lsmeans R package (v2.30–0; *Lenth, 2016*) to identify a significant difference in the expression level of genes marking different cell types between different muscles. We used the pheatmap function from the pheatmap R package (v1.0.12; https://CRAN.R-project.org/package=pheatmap) with the difficult setting to draw all the heatmaps.

## Differential expression analysis (DEA)

We used the *voom*-transformed data to fit linear mixed-effects models for each gene using the lmer function from the lmerTest R package. The individual and muscle were included in the models as a random-effect and a fixed-effect, respectively, similarly to the formula-2. The *voom* precision weights showing the mean-variance trend for each observation were incorporated into the models. We tested the significance of fixed effects with the ANOVA from the car R package and the FDR was applied to adjust for multiple testing. We conducted post-hoc pairwise comparisons using the lsmeans R package to identify significant differences between each pair of muscles.

We calculated the Pearson correlation between differentially expressed genes (DEGs, FDR <0.05) and the eigenvector of each cell type using the cor and cor.test from the stats R package. We adjusted for multiple testing using the FDR. DEGs which were significantly associated with a cell type eigenvector (Pearson correlation >0.5 and FDR <0.05) were defined as cell type related.

To find differentially expressed transcripts (DETs), we performed the DEA using the same model (formula-2) for the voom-transformed transcript data. We defined cell type related DETs as transcripts originating from genes with significant association with a cell type eigenvector.

## Differential transcript usage analysis

We computed the isoform usage differences using IsoformSwitchAnalyzeR Bioconductor package (v1.20.0; *Vitting-Seerup and Sandelin, 2019*). Breifly, the Ballgown input table files generated by StringTie were imported by the importIsoformExpression function. A switchAnalyzeRlist was created considering the muscle and batch in the design matrix using importRdata function. The transcripts of genes that were not present in the filtered gene dataset were filtered out using subsetSwitchAnalyzeRlist function. Finally, isoformSwitchAnalysisCombined function was used to determine splice switch changes (differential transcript usage) between each pair of muscles. The significant splice switch changes were identified using significant FDR (<0.05).

## Consensus gene co-expression network analysis

In order to construct a gene network, we used the weighted gene co-expression network analysis algorithm using the WGCNA R package (v1.69) (*Langfelder and Horvath, 2008*). We used the *voom* transformed data as an input. In order to calibrate the parameters of the network, we used the approach published by our group (*Abbassi-Daloii et al., 2020*). Briefly, prior knowledge of gene interactions from a pathway database was used to select the most optimal set of WGCNA parameters. We used the biweight midcorrelation (median-based) function in WGCNA of the signed hybrid type to define the adjacency matrix. We performed a full parameter sweep, testing various combinations of settings for power (6, 8, 10, 12, 14, 18, and 22), minClusterSize (15, 20, and 30), deepSplit (0, 2, and 4), and

CutHeight (0.1, 0.15, 0.2, 0.25, and 0.3). These different settings were assessed using the knowledge network obtained from the Reactome database using g:ProfileR2 R package (v0.2.0) (**Kolberg et al., 2020**). All possible pairs of genes were assigned into four different groups: (1) in the same module and in the same pathway, (2) in the same module but not in the same pathway, (3) not in the same module but in the same pathway and (4) neither in the same module nor in the same pathway. The enrichment factor ($\frac{No.pairs \in group1 \times No.pairs \in group4}{No.pairs \in group2 \times No.pairs \in group3}$) was calculated. The optimal set of parameters with the highest enrichment factor was: power: 8, MinModuleSize: 20, deepSplit: 0, Cut Height: 0.2. To identify gene co-expression networks that were consistent across individuals, we constructed first co-expression networks for each individual separately and merged these subsequently into a consensus co-expression network. To achieve this, the adjacency matrices per individual were raised to power 8 and converted into topological overlap matrices (TOM). TOM of some individuals may be overall lower or higher than TOM of other individuals. To account for this, we performed percentile (0.95) normalization over all the TOMs. The consensus TOM was then calculated by taking the elementwise 40th percentile of the TOMs. The consensus TOM was used to calculate the TOM dissimilarity matrix ($dissTOM = 1 - TOM$) which was then input to agglomerative hierarchical clustering (**Langfelder and Horvath, 2012**). Finally, modules were identified using a dynamic tree-cutting algorithm from the resulting dendrogram (**Langfelder et al., 2008**) specifying MinModuleSize =20 and deepSplit =0. The module labeled 'gray' was not considered in the analysis as it consisted of genes that did not assign to any specific module. The summary expression measure for each module, the module eigengene (ME), was calculated (**Zhang and Horvath, 2005**). Modules with similar expression profiles were merged at the threshold of 0.2. In addition, we calculated the intramodular connectivity to identify highly interconnected genes, called hub genes, per module.

## Module-muscle association

To identify modules that differ in expression levels between muscles (named as muscle-related modules), we fitted linear mixed-effect models on the module eigengenes (MEs) using the lmer function from the lmerTest R package. These models included individual as a random-effect and muscle as a fixed-effect, similar to formula-2. We tested the significance of fixed effects with ANOVA from the car R package. We used ranova from the LmerTest R package to test the significance of random effects. To identify significant differences between each pair of muscles, we used a post-hoc multiple comparison tests as implemented in the lsmeans R package.

We performed a functional enrichment analysis for the muscle-related modules using ClueGO App (v2.5.7) (**Bindea et al., 2009**) in Cytoscape (v3.8.1) (**Kohl et al., 2011**). We used the CyREST API (**Ono et al., 2015**) to execute the ClueGO by R script (http://www.ici.upmc.fr/cluego/cluegoDocumentation.shtml). Pathways and gene annotations from Kyoto Encyclopedia of Genes and Genomes (KEGG), Gene Ontology (GO), Reactome, and WikiPathways (WP) were included. The Benjamini-Hochberg FDR was applied to adjust for multiple testing. The annotations with any differentially expressed genes or hub genes or a transcription factor were included. To eliminate the redundant annotations, we only included an annotation with the lowest FDR for each 'GoGroups' defined by ClueGO and the annotations marked as 'LeadingGoTerm' by ClueGO.

We next determined muscle-related modules which showed the largest differences between the three groups of muscles. These modules were selected based on the FDR of GlueGO enrichment (<0.01) and the F-value of the genes resulting from DEA in each module (third quantile >5.5).

## Immunofluorescence staining, imaging, image analysis

The immunofluorescence staining included myofiber typing and capillary staining. Prior to the staining, slides were allowed to equilibrate to room temperature, blocked for 30 min using 5% milk powder (FrieslandCampina, Amersfoort, The Netherlands) in phosphate-buffered saline containing 0.05% tween (PBST).

## Myofiber type composition

The myofiber immunostaining, image quantification and myofiber type composition analysis are detailed in a STAR protocol (**Abbassi-Daloii et al., 2023**).

## Myosin staining

The antibodies for three myosin heavy chain (MyHC) isoforms (MyHC1, MyHC2A, and MyHC2X) and laminin were used as described by *Riaz et al., 2016*. Briefly, cryosections were stained with rabbit anti-laminin (1:1000, Sigma-Aldrich, L9393) and mouse anti-6H1 (1:5, DSHB; AB_2314830) detecting MyHC2X, for two hours at room temperature. Following the PBST washing, the secondary antibodies goat anti-rabbit-conjugated-Alexa Fluor 750 (1:1000, Thermo Fisher Scientific, A21039) and goat anti-mouse-conjugated-Alexa Fluor 488 (1:1000, A11001, Thermo Fisher Scientific) were incubated for an hour at room temperature. After PBST washing, sections were incubated overnight at four degrees with a mix of fluorescently conjugated monoclonal antibodies: BA-D5-conjugated-Alexa Fluor 350 (1:600, DSHB, AB_2235587) and SC-71-conjugated-Alexa Fluor 594 (1:700, DSHB, AB_2147165), detecting MyHC1 and MyHC2A, respectively. Lastly, after washing with PBST, the cryosections were mounted with ProLong Gold antifade reagent (P36930, Thermo Fisher Scientific) and stored at four degrees prior to imaging.

## Image acquisition, processing, and quantification

The stained slides were imaged with the Axio Scan.Z1 slidescanner (Carl Zeiss, Germany) using the ZEN Blue software (v2.6), capturing the entire section. The images were acquired with a 10×/0.45 Plan-Apochromat objective lens and the same image settings were used for all slides.

After imaging all cryosections, a shading profile was calculated using the 'Shading Reference From Tile Image' in ZEN Lite (v3.3) for each channel in each slide. This procedure produces a shading profile for each channel per slide and does not apply the shading correction. To improve the accuracy of the shading profile, we calculated the median over all the shading profiles over all scanned slides for each channel. These median shading profiles were then used to perform the shading correction using 'Shading Correction' in ZEN Lite (v3.3).

Further image processing was performed using Fiji (v 1.51) (*Schindelin et al., 2012*). Since the aggregated dataset is relatively large, we created a modular set of Fiji macros that process each step independently.

First, we converted the slidescanner datasets from the Carl Zeiss Image format (CZI) to multi-channel 16 bit TIFF files using BioFormats (*Linkert et al., 2010*). In this step, the images were 4 x downsampled, by averaging, to improve the processing speed and reduce the dataset size. After downsampling the effective pixel size was 2.6 µm.

Next, we applied a semi-automated process to generate tissue masks from the laminin channel to determine the (parts of) cryosections to be quantified. To generate masks, we first used an automated procedure, inspired by 'ArtefactDetectionOnLaminin' method from MuscleJ (*Mayeuf-Louchart et al., 2018*). Subsequently, a manual step was incorporated to check and correct the generated masks. For each sample, we performed manual corrections to remove artifacts such as tissue folds, out-of-focus regions, scratch, and dirt objects.

Then, we generated 'masked' copies of the laminin channel. To reduce any possible artifacts due to this binary mask, we applied a gaussian blur of 4 pixels to the masks and we set the pixel values of the laminin channel that were outside the mask to the median intensity of these pixels. The masked laminin images were then fed into the *Ilastik* pixel classification algorithm (*Berg et al., 2019*). In *Ilastik* we trained a classifier to identify two classes: 'myofiber boundary' and 'not myofiber boundary'. This classifier was then used to process all images in this dataset. This classification step greatly improved the subsequent laminin segmentation outputs.

Next, the laminin objects were segmented based on the output of the previous step. In short, the image was slightly blurred with a Gaussian Blur, after which the image was segmented using the Fiji method 'Find Maxima' with output 'Segmented Particles', followed by binary dilation, and closing. Finally, the regions-of-interest (ROI; individual laminin segmented objects) were generated using the 'Analyze Particle' method from Fiji.

After laminin segmentation, we measured the mean fluorescence intensity (MFI) as well as other properties in ROIs for all three channels using the Fiji measurement: 'Mean gray value'. In addition, we recorded the 'Area', 'Standard deviation', 'Modal gray value', 'Min & max gray value', 'Shape descriptors', and 'Median' features. We also quantified the results of the pixel-classification step by measuring its 'Mean gray value' in each ROI as well as on the border (3-pixel enlargement) of each ROI. This quantification allows the assessment of the myofiber 'segmentation certainty', the certainty

is high when the pixel-classification is high for the 'myofiber boundary' class all around the myofiber and low in the interior of the myofiber.

## Myofiber type composition analysis

First, we filtered out the non-myofiber objects since the laminin segmentation was automatic. We applied a percentile filtering for a 'segmentation certainty' on the cross-sectional area (CSA) and the circularity values. The objects with (**1**) pixel-classification on the object boundary less than 5th percentile or (**2**) pixel-classification in the interior of the object greater than 95th percentile or (**3**) CSA less than 10th percentile or greater than 99th percentile or (**4**) circularity greater than 1st percentile were excluded. Samples from different muscles were pooled for all different filtering criteria except for the filtering for CSA, as the density distributions of CSA were found to differ between different muscles. In the next step, we selected the cryosection with the largest number of myofibers for each sample for further analysis. Samples with a minimum of a hundred myofibers were included in the myofiber type analysis. The final dataset contained 1,287,729 myofibers from 96 samples, with a median of 888 myofibers per sample. As previously described by *Raz et al., 2020*, per myofiber, the MFI values for each of three MyHC isoforms were scaled per sample (without centering). Subsequently, the composition of myofiber types was determined by clustering of the transformed (natural logarithm) MFI values. Each myofiber was assigned to a cluster using the mean-shift algorithm (bandwidth ($h$)=0.02), a density-based clustering approach, implemented in the LPCM R package (v0.46–7) (*Cheng, 1995*; *Durham University and Einbeck, 2011*). All the small clusters, with less than 1% from the total myofibers, were excluded. Then, per myofiber type cluster, the proportions of the total myofibers were calculated per sample.

## Capillary density

### Staining and image acquisition

Sections were stained with the primary antibodies: anti-human CD105 (endoglin, ENG) biotin-conjugated (1:100, BioLegend, 323214), anti-human CD31-Alexa Fluor 594 conjugated (1:400, BioLegend, 303126), and rabbit anti-laminin for two hours at room temperature. After PBST washing, the slides were incubated with streptavidin-Alexa Fluor 647 conjugated (1:500, Life Technologies, S21374) and goat anti-rabbit Alexa Fluor 750-conjugated for an hour. After final PSBT washing, nuclei were counterstained with 4',6-diamidino-2-phenylindole (DAPI) (0.5 µg/mL, Sigma-Aldrich) and were mounted with ProLong Gold antifade reagent. Cryosections were imaged with Axio Scan.Z1 slide scanner.

### Image processing and quantification

We used Fiji macros created for the myofiber composition analysis to convert CZI files to TIFF files, to generate the masks, and for the laminin segmentation. We then measured the cross-sectional area for laminin segmented objects using the Fiji 'Area' measurement. Next, a Gaussian Blur filter with an σ value set to 1 was implemented on the CD31 channel, followed by thresholding using setAuto-Threshold ("Li dark" algorithm) and processing using Watershed algorithm to separate touching and overlapping cells. The lumens were filled using the Fill Holes algorithm in Fiji. We then measured the properties in ROIs using the Fiji measurements: 'Area', 'Mean gray value', 'Standard deviation', and 'Shape descriptors'. We then implemented the same processing on the ENG channel to select the ROIs but measured the 'Mean gray value' and 'Standard deviation' in the CD31 channel to determine the CD31 and ENG colocalization.

For the image quantification, we first calculated the ratio between the total positively stained areas for CD31 and the total area of the muscle section, expressed as a percentage. We then determined the capillaries as the objects with (1) positive signals for both CD31 and ENG (*Wehrhan et al., 2011*), (2) larger than 3 µm$^2$ and smaller than 51 µm$^2$ (*Poole et al., 2013*), and (3) circularity larger than 0.5. Finally, we defined capillary density as the number of capillaries per unit (µm²) of muscle area.

## RNAscope in situ hybridization

We detected single-molecule RNA using Multiplex Fluorescent Reagent Kit v2 (ACDBio, 323135) according to the manufacturer's protocol for fresh-frozen cryosections, with the following adjustments to optimize the experiment for human muscles: fixation with 4% paraformaldehyde at 4 degrees for

an hour, and all washing steps with washing buffer were performed three times for 2 min each. The protocol was optimized on control muscle cryosections by negative and positive probe sets provided by ACDBio. *Figure 8—figure supplement 1* shows a representative image of negative control probes in a muscle cryosection. We performed the hybridization using probes for *Hs-HOXA11* (ACDBio, 1061891-C1), *Hs-HOXA10* (ACDBio, 867141-C2), and *Hs-HOXC10* (ACDBio, 803141-C3). Following the completion of the RNA probe hybridization, we carried out an immunostaining step at room temperature to label myofibers with rabbit anti-laminin followed by secondary labeling with goat anti-rabbit-conjugated-Alexa Fluor 555 (1:1000, Abcam, ab150078). Lastly, following PBST washing, the nuclei were counterstained with DAPI (ACDbio, 323110). Cryosections were mounted with ProLong Gold antifade reagent. Slides were imaged with a Leica SP8 confocal microscope, equipped with a white light laser (WLL) source (Leica Microsystems, Germany) using a 40 x/1.3 OIL objective. For each sample, multiple tiles at different regions across the muscle cryosection were images with seven z-planes (z-step size = 0.35 μm). The images for DAPI and *HOXA11* channels were acquired using a HyD 2 detector with 414 nm-532nm excitation lasers and with 504 nm-543nm excitation lasers, respectively. A HyD 4 detector was used to image anti-laminin and *HOXA10* channels with 558 nm-585nm excitation lasers and with 603 nm-665nm excitation lasers, respectively. A HyD 5 detector was used to image *HOXC10* channel with 675 nm-800nm excitation lasers. The same image settings were used for all samples.

We performed the image processing in multiple steps and created a modular set of Fiji macros that process each step independently. We first merged and converted the Leica Image File (LIF) to a multichannel 16 bit TIFF file using the Grid/Collection Stitching Plugin (*Preibisch et al., 2009*). We segmented myofibers using the following steps: (1) creating the maximum intensities projections of the laminin channel, (2) creating 'probability' maps of the laminin channel in *Ilastik*, (3) adding a point selection to the TIFF files, which seed the watershed, and (4) implementing watershed segmentation with two halting points for user interaction, first watershed segmentation and then making the ROI list (individual segmented myofibers) generated using the 'Analyze Particle' command.

After myofiber segmentation, we implemented a Gaussian Blur filter with an σ value set to 1 on each probe channel. We then applied the color threshold settings using setAutoThreshold ('RenyiEntropy dark' algorithm). Finally, for each probe channel, we measured the foci properties in each segmented myofiber using the Fiji measurements: 'Area', 'Mean gray value', 'Standard deviation', and 'Shape descriptors'.

The RNA foci were defined as speckles smaller than 3.5 μm$^2$ with circularity above 0.98. We excluded *HOXA11* from further analysis due to a low signal-to-noise ratio. Based on the negative controls, we defined threshold values to filter out false-positive signals for the 2 other HOX genes. These threshold values were set such that approximately all the foci in the negative control were classified as negative. Finally, to compare the expression of two genes between muscles, we calculated the average number of foci per myofiber per sample.

## Availability of data and scripts

All scripts are publicly available on GitHub: https://github.com/tabbassidaloii/HumanMuscleTranscriptomeAtlasAnalyses, (*Abbassi-Daloii, 2023* copy archived at swh:1:rev:4d885b7612c66147c99a26c-4503559deabc610ac). The raw data is publicly available at the European Genome Archive (Dataset ID: EGAS00001005904, https://ega-archive.org). *Figure 1C* and *Figure 1—figure supplement 1* show our analyses workflow used to explore genes contributing to the intrinsic differences between muscles.

## Graphical user interface

The muscle transcriptomics atlas is available for exploration through a graphical user interface (https://tabbassidaloii.shinyapps.io/muscleAtlasShinyApp/) implemented using shiny, a web application framework for application shiny R package (v1.5.0)(*Chang et al., 2020*).

## Gene network visualization

The subnetwork was exported and visualized in Cytoscape (v3.8.1).

## Acknowledgements

We thank Susan Kloet, and the personnel from the Leiden genome technology center (LGTC) in the LUMC for providing the sequencing support. HK is member of the European Reference Network for Rare Neuromuscular Diseases [ERN EURO-NMD].

This project was funded by the Netherlands Organization for Scientific Research (NWO, under research program VIDI, Grant # 917.164.90) and the Association Française contre les Myopathies (AFM Telethon; Grant # 22506). We thank the personnel of the Sequence Analysis Support Core (SASC) in the LUMC for their support in data pre-processing and data submission at the European Genome Archive. This work was partially funded by a grant to the Netherlands X-omics Initiative (Dutch Research Council (NWO), project 184.034.019)

## Additional information

### Funding

| Funder | Grant reference number | Author |
|---|---|---|
| Nederlandse Organisatie voor Wetenschappelijk Onderzoek | 917.164.90 | Hermien E Kan |
| Association France Myopathies | 22506 | Vered Raz |
| Netherlands X-omics Initiative | 184.034.019 | Peter AC 't Hoen |

The funders had no role in study design, data collection and interpretation, or the decision to submit the work for publication.

### Author contributions

Tooba Abbassi-Daloii, Resources, Data curation, Formal analysis, Methodology, Writing – original draft; Salma el Abdellaoui, Visualization; Lenard M Voortman, Software, Visualization, Methodology; Thom TJ Veeger, Writing – review and editing; Davy Cats, Software; Hailiang Mei, Data curation; Duncan E Meuffels, Ewoud van Arkel, Resources; Peter AC 't Hoen, Conceptualization, Resources, Software, Supervision, Methodology; Hermien E Kan, Conceptualization, Supervision, Funding acquisition, Project administration, Writing – review and editing; Vered Raz, Conceptualization, Supervision, Funding acquisition, Methodology, Project administration, Writing – review and editing

### Author ORCIDs

Tooba Abbassi-Daloii http://orcid.org/0000-0002-4904-3269
Lenard M Voortman http://orcid.org/0000-0001-9794-067X
Duncan E Meuffels http://orcid.org/0000-0002-5372-6003
Peter AC 't Hoen http://orcid.org/0000-0003-4450-3112
Hermien E Kan http://orcid.org/0000-0002-5772-7177
Vered Raz http://orcid.org/0000-0003-3152-1952

### Ethics

The study was approved by the local Medical Ethical Review Board of The Hague Zuid-West and the Erasmus Medical Centre and conducted in accordance with the ethical standards stated in the 1964 Declaration of Helsinki and its later amendments (ABR number: NL54081.098.16). All subjects provided written informed consent prior to participation.

### Decision letter and Author response

Decision letter https://doi.org/10.7554/eLife.80500.sa1
Author response https://doi.org/10.7554/eLife.80500.sa2

# Additional files

### Supplementary files
- Supplementary file 1. Samples' metadata.
- Supplementary file 2. A combined list of the genes marking different cell types.
- Supplementary file 3. The result of differential expression analysis for genes.
- Supplementary file 4. The result of differential expression analysis for transcripts.
- Supplementary file 5. The result of differential transcript usage analysis.
- Supplementary file 6. List of genes – modules.
- Supplementary file 7. List of enriched biological processes and molecular functions within modules.
- MDAR checklist

### Data availability
The raw data is publicly available at the European Genome Archive (Dataset ID: EGAS00001005904, https://ega-archive.org/). The muscle transcriptomics atlas is available for exploration through a graphical user interface (https://tabbassidaloii.shinyapps.io/muscleAtlasShinyApp/).

The following previously published datasets were used:

| Author(s) | Year | Dataset title | Dataset URL | Database and Identifier |
|---|---|---|---|---|
| Abbassi-Daloii T | 2022 | The raw data is publicly available at the European Genome Archive | https://ega-archive.org/ | European Genome Archive, EGAS00001005904 |

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
