## [Editor Report]

Skeletal muscle groups varied in biomechanical and can show a diffential involvement pattren in muscular disorders. Molecular characteristics of leg muscles from healthy young human males is presented in this study. It provides the first comparitive of gene signatures that will be valuable to understand muscle involvement in normal and abnormal physiological and pathological conditions.

---

## [Decision Letter]

**Decision letter after peer review:**

Thank you for submitting your article "A transcriptome atlas of leg muscles from healthy human volunteers reveals molecular and cellular signatures associated with muscle location" for consideration by *eLife*. Your article has been reviewed by 2 peer reviewers, and the evaluation has been overseen by a Reviewing Editor and Mone Zaidi as the Senior Editor. The reviewers have opted to remain anonymous.

Essential revisions:

A) Points raised by reviewer 1:

– The differences in functions because of different proportions of slow and fast fibers are well known and self evident. Because of the large interindividual variability which was greater than the variations between the different muscles under study the authors should discuss more how to methodologically enable higher levels of distinction between the different muscles.

– Analyzing the observed expression differences according to genetic myopathy gene defect categories such as: sarcolemmal genes, dystroglycanopathy genes, myofibrillar genes, autophagy related genes, chaperones, ion channels etc. would have been a preferred approach as to understand the differences and sometimes very selective involvement of certain muscles in the different genetic muscle diseases

– Instead of comparing 5 thigh muscles and one lower leg muscle maybe comparing six different lower leg muscles could have yielded more detail?

– The separation of different fast-slow fiber types could be further detailed by using the double IHC staining technique which identifies correctly all specific type I, IIA and IIX fibers and also the IIA/IIX hybrid fibers and the I/IIA hybrids for quantification.

– Another aspect not discussed would be the expression of different isoforms of the muscle genes which could have major influence on the functional profiles and the susceptibility or resistance to degeneration with certain gene defects.

B) Points raised by reviewer 2:

These data provide an impressive dataset of human muscle gene expression from young healthy individuals and will serve as a reference for future investigation of muscle pathology or ageing. The conclusions of this paper are mostly well supported by the data, but some aspects of the analyses by immunohistochemistry need to be clarified.

1) The method to cluster fibres according to their relative expression of myosin isoforms needs to be further explained (Figure 3). Fibres were clustered into three main categories according to the fluorescence intensity of MyHC1, MyHC2A and MyHC2X after immunohistochemistry. Figure 3B shows a continuum of the three intensities rather than clearly separated clusters and the authors should explain how fluorescence intensity thresholds were used to define such clusters. In addition, the differences in average Mean Fluorescence Intensity (MFI) values between clusters appear quite low: Max(MyHC1)/Min(MyHC1)=0.725/0.653=1.11;

Max(MyH2A)/Min(MyH2A)=0.806/0.535=1.50;

Max(MyH2X)/Min(MyH2X)=0.718/0.651=1.10.

These differences are low, notably compared to the gene expression data where these myosin isoforms have expression levels spread over 2 logs (Figures 3D-F). The authors should show specific examples of quantification, show specific quantification of fluorescence over MFI of background, and add statistics of expression differences based on MFI for each MyHC between clusters.

Figures 3D-F show correlations between the % of fibres in a given fluorescence cluster as a function of the expression by RNAseq of the corresponding MyHC defining this cluster. To strengthen these correlations, the authors should show the % of fibres in a given fluorescence cluster as a function of the expression of all three MyHC isoforms (e.g. % of fibres in cluster 1 = f(expression MYH7) or f(expression MYH1), same for clusters 2 and 3).

2) The authors assess blood vessel density in GL by immunohistochemistry (Figure 4). Figure 4A does not mention the muscle corresponding to the cryosection presented, and presenting STM and GL sections side-by-side would help to understand the conclusions of this figure. Displaying the green and red arrows as well on the CD31 and ENG panels and showing higher magnifications would help to understand which regions were defined or not as capillaries.

3) The authors validate HOX expression patterns by RNA-scope (Figure 8). Figure 8A does not mention the muscle corresponding to the cryosection presented. Maybe it would help show STM and GL sections stained for HOXA10 and HOXC10 side-by-side. Also, although the authors mention that signal specificity of RNA-scope probes was verified using negative controls, it would be helpful to show these controls and validate these probes on muscle sections known not to express HOX genes (head-derived?).

Finally, gene expression datasets from human muscle samples have already been generated. As discussed by the authors, these studies were limited in terms of the number of samples, large variation of donor ages, sample conservation before processing, etc. Nevertheless, it would be helpful to put the main findings of this paper (cell type composition, blood vessel density, fibre types ratios, HOX genes expression, mitochondrial processes, etc) into context and assess, if possible and if the data is available, whether similar findings can be concluded from previous datasets.

---

## [Author Response]

Essential revisions:A) Points raised by reviewer 1:1) The differences in functions because of different proportions of slow and fast fibers are well known and self-evident. Because of the large interindividual variability which was greater than the variations between the different muscles under study the authors should discuss more how to methodologically enable higher levels of distinction between the different muscles. Instead of comparing 5 thigh muscles and one lower leg muscle maybe comparing six-even different lower leg muscles could have yielded more detail?

We address this comment in two parts:

– Methodology to overcome inter individual differences:

In the revised version we extended the discussion how we dealt with the inter individual variability to optimize the discovery of genes with different expression levels between muscles (page 9, lines 337-350). We added sentences on how we accounted for the differences between individuals in the statistical models we employed and in the consensus network. In brief, in both cell type analysis and differential expression analyses, we included individuals as a random effect to the statistical model. In the co-expression network analysis (WGCNA), a network was constructed for each individual followed by a consensus network construction to correct for the individual’s effect. In the downstream steps of WGCNA, the same as the models above, individuals were included as a random effect.

– Muscles inclusion:

The muscles in this study are all leg muscles, but in terms of their role in locomotion, the muscles above or under the knee and the hamstrings and quadriceps have different roles (Willigenburg, McNally et al. 2014, Camic, Kovacs et al. 2015).

Muscle inclusion was determined by the surgeons of the ACL surgery. We included as many muscles as possible.

2) Another aspect not discussed would be the expression of different isoforms of the muscle genes which could have a major influence on the functional profiles and the susceptibility or resistance to degeneration with certain gene defects.

In our initial submission, we focused on genes’ analysis (combining multiple transcripts per gene). Following the reviewer's suggestion, we now added transcript-level differential expression analysis and transcript usage analysis: we quantified transcript expression levels using StringTie. After batch effect correction (the same approach used for the gene expression count table), we performed differential expression analysis. The approach is detailed in the material and method section (page 14, lines 535-538). The results are shown in Figure 5B.

Briefly, the results showed that the muscles clustered into the same groups as genes, considering the proportion of transcript with differential expression (DETs) in each pair of muscles. One panel has been added to Figure 5 and one to Figure 5 —figure supplement 1.

Collectively, these results imply that not only transcriptional but also post-transcriptional mechanisms contribute to the intrinsic differences between different leg muscles.

B) Points raised by reviewer 2:These data provide an impressive dataset of human muscle gene expression from young healthy individuals and will serve as a reference for future investigation of muscle pathology or ageing. The conclusions of this paper are mostly well supported by the data, but some aspects of the analyses by immunohistochemistry need to be clarified.

We appreciate this positive evaluation of our resources.

1) The method to cluster fibres according to their relative expression of myosin isoforms needs to be further explained (Figure 3). Fibres were clustered into three main categories according to the fluorescence intensity of MyHC1, MyHC2A and MyHC2X after immunohistochemistry. Figure 3B shows a continuum of the three intensities rather than clearly separated clusters and the authors should explain how fluorescence intensity thresholds were used to define such clusters. In addition, the differences in average Mean Fluorescence Intensity (MFI) values between clusters appear quite low:Max(MyHC1)/Min(MyHC1)=0.725/0.653=1.11;Max(MyH2A)/Min(MyH2A)=0.806/0.535=1.50;Max(MyH2X)/Min(MyH2X)=0.718/0.651=1.10.These differences are low, notably compared to the gene expression data where these myosin isoforms have expression levels spread over 2 logs (Figures 3D-F). The authors should show specific examples of quantification, show specific quantification of fluorescence over MFI of background, and add statistics of expression differences based on MFI for each MyHC between clusters.

A detailed protocol for myofiber typing, wet lab, computer analysis (including clustering) and examples of MFI quantification can be found in a manuscript that we recently published: https://doi.org/10.5281/zenodo.7186929. This paper has been accepted for publication in STAR protocols and it is cited in the text (page 16, lines 613-615).

To find the myofiber clusters, we used the meanshift method, an unsupervised approach that clusters the myofibers based on the MyHC MFI. In this method, there is no MFI threshold, but clustering is made by the bandwidth value, i.e the distance/size scale of the kernel function. In other words, the size of the “window” used to calculate the mean.

We agree that using manual imaging background correction should be applied, also to eliminate differences between imaging sessions. We always applied it in our previous studies. In our study, imaging was carried out with a slide scanner in one batch. Since we obtained myofibers negative for each one of the fluorophores there was not need for background correction. The MFI of all three MyHC isoforms were scaled (without centering) and transformed prior to calculations or clustering. This is now specifically indicated in the legend to figure 3C. In our studies we consider the distribution of MFI values instead of the mean calculated by min and max values as suggested by the reviewer. Our approach does not assume normal distribution of the dataset, which is often the case in imaging dataset.

We have not implemented any statistics comparing the MFI for each MyHC isoform between clusters. However, the choice of bandwidth value is arbitrary and will affect the number and the size of clusters. Therefore, we selected the bandwidth where the number of myofiber clusters was biologically relevant (e.g. choosing a bandwidth that resulted in 10 myofiber clusters would not be biologically relevant).

As correctly spotted by the reviewer, Figure 3B (clustering 3D plot) shows a large variation in each cluster. As a result, the mean MFI values in the table (Figure 3C) are not representative of the cluster. Therefore, we replaced the table with density plots (one for each cluster) showing MFI distribution for each MyHC isoform (new Panel 3C). We believe this presentation is clearer.

Figures 3D-F show correlations between the % of fibres in a given fluorescence cluster as a function of the expression by RNAseq of the corresponding MyHC defining this cluster. To strengthen these correlations, the authors should show the % of fibres in a given fluorescence cluster as a function of the expression of all three MyHC isoforms (e.g. % of fibres in cluster 1 = f(expression MYH7) or f(expression MYH1), same for clusters 2 and 3).

As suggested by reviewer, we included a panel to Figure 3 —figure supplement 1A, showing the proportion of myofibers in each cluster as a function of the expression of all three MyHC isoforms. The additional panel confirms that the observed differences in *MYH* RNA expression are consistent with differences in the myofiber compositions and the assignments of cluster 1 as predominantly type 2A, cluster 2 as predominantly type 1 and cluster as predominantly 2X.

2) The authors assess blood vessel density in GL by immunohistochemistry (Figure 4). Figure 4A does not mention the muscle corresponding to the cryosection presented, and presenting STM and GL sections side-by-side would help to understand the conclusions of this figure. Displaying the green and red arrows as well on the CD31 and ENG panels and showing higher magnifications would help to understand which regions were defined or not as capillaries.

The figure is updated based on the reviewer recommendation:

3) The authors validate HOX expression patterns by RNA-scope (Figure 8). Figure 8A does not mention the muscle corresponding to the cryosection presented. Maybe it would help show STM and GL sections stained for HOXA10 and HOXC10 side-by-side. Also, although the authors mention that signal specificity of RNA-scope probes was verified using negative controls, it would be helpful to show these controls and validate these probes on muscle sections known not to express HOX genes (head-derived?).

We added an image zoom-into a single myofiber from each muscle.

Unfortunately, since this procedure is highly costly, we did not stain sections from muscles with no expression of these genes. We included the images from a negative control, where the muscle was stained with a bacterial mRNA probe (Figure 8 —figure supplement 1),

4) Finally, gene expression datasets from human muscle samples have already been generated. As discussed by the authors, these studies were limited in terms of the number of samples, large variation of donor ages, sample conservation before processing, etc. Nevertheless, it would be helpful to put the main findings of this paper (cell type composition, blood vessel density, fibre types ratios, HOX genes expression, mitochondrial processes, etc) into context and assess, if possible and if the data is available, whether similar findings can be concluded from previous datasets.

To our knowledge, this is the first study comparing gene expression patterns from human 6 leg muscles in a single study. Most gene expression studies comparing different muscles have been performed in mice. In the discussion, we thus compared our results to earlier immunohistochemistry-based studies in humans on the fiber type composition and capillary density. Moreover, we put our results on *HOX* gene expression in the context of earlier mouse gene expression studies (page 8, lines 289-304).